



# In-orbit results of the Coupled Dark State Magnetometer aboard the China Seismo-Electromagnetic Satellite

Andreas Pollinger[1,2], Christoph Amtmann[2], Alexander Betzler[2,1], Bingjun Cheng[3], Michaela Ellmeier[1,2], Christian Hagen[2,1], Irmgard Jernej[1], Roland Lammegger[2], Bin Zhou[3], Werner Magnes[1]

[1]Space Research Institute, Austrian Academy of Sciences, Graz, 8042, Austria
[2]Institute of Experimental Physics, Graz University of Technology, Graz, 8010, Austria
[3]National Space Science Center, Chinese Academy of Sciences, Beijing, 100190, China

*Correspondence to*: Andreas Pollinger (andreas.pollinger@oeaw.ac.at)

**Abstract.** The China Seismo-Electromagnetic Satellite (CSES) was launched in February 2018 into a polar, sun-synchronous,
low Earth orbit. It provides the first demonstration of the Coupled Dark State Magnetometer (CDSM) measurement principle
in space. The CDSM is an optical scalar magnetometer based on the Coherent Population Trapping (CPT) effect and measures
the scalar field with the lowest absolute error aboard CSES. Therefore, it serves as the reference instrument for the
measurements done by the fluxgate sensors within the High Precision Magnetometer instrument package.

In this paper several correction steps are discussed in order to improve the accuracy of the CDSM data. This includes the
extraction of valid 1 Hz data, the application of the sensor heading characteristic, the handling of discontinuities at CPT
resonance transitions as well as the removal of fluxgate and satellite interferences.

The in-orbit performance is compared to the Absolute Scalar Magnetometer aboard the SWARM satellite Bravo via the
CHAOS-6 magnetic field model. Additionally, an uncertainty of the magnetic field measurement is derived from unexpected
parametric changes of the CDSM in orbit in combination with performance measurements on ground.

## 1 Introduction

The China Seismo-Electromagnetic Satellite (CSES), also known as Zhangheng-1, investigates natural electromagnetic
phenomena and possible applications for earthquake monitoring from space (Shen et al., 2018). CSES was launched in
February 2018 into a polar, sun-synchronous, low Earth orbit with an inclination of approx. 97° and a period of approx. 95
minutes. The High Precision Magnetometer (HPM) instrument package (Cheng et al., 2018) consists of two FluxGate
Magnetometers (FGMs) in a gradiometer configuration and the Coupled Dark State Magnetometer (CDSM). The CDSM
measures the scalar field with the lowest absolute error of the instruments aboard CSES and serves as the reference instrument
for the measurements done by the fluxgate sensors.

The CDSM is an optical scalar magnetometer based on a quantum interference effect called Coherent Population Trapping
(CPT) (Arimondo, 1996; Wynands and Nagel, 1999) which inherently enables omni-directional measurements (Lammegger,


2008; Pollinger et al., 2012) and an all-optical sensor design without double cell units, excitation coils or active electronics parts (Pollinger et al., 2018).

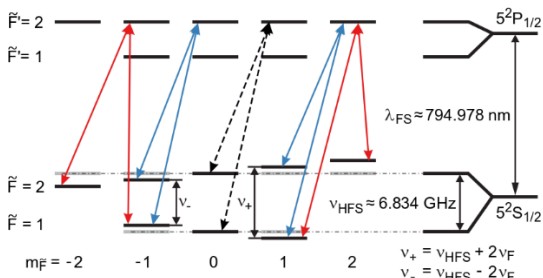

**Figure 1: Laser excitation scheme within the D$_1$-line hyperfine structure of $^{87}$Rb.**

The instrument simultaneously probes several CPT resonances which are established within the D$_1$-line HyperFine Structure
(HFS) of $^{87}$Rb shown in Fig. 1. Here, the total angular momentum quantum numbers and the magnetic quantum numbers of the $5^2$S$_{1/2}$ ground states are denoted by $\tilde{F}$ and, correspondingly, by $m_{\tilde{F}}$ while for the $5^2$P$_{1/2}$ excited states the labels are primed. The wavelength λ$_{FS}$ corresponds to the fine structure transition $5^2$S$_{1/2}$ → $5^2$P$_{1/2}$. The HFS ground state splitting frequency is denoted by ν$_{HFS}$. The energy shift introduced by the magnetic field is expressed by ν$_F$. For the excitation of each CPT resonance, a Λ-shaped excitation scheme is prepared in the HFS which consists of three energy levels interacting with two light fields,
which are indicated by the arrowed lines in Fig. 1.

In order to create the necessary light fields, a Vertical-Cavity Surface-Emitting Laser (VCSEL) diode (vacuum wavelength λ$_{Laser}$ = λ$_{FS}$ ≈ 794.978 nm) is Frequency Modulated (FM) by a microwave oscillator signal (f$_{MW}$ = ½ ν$_{HFS}$ ≈ 3.417 GHz). The CPT resonance n = 0 is excited by the two light fields denoted by the black dashed arrowed lines in Fig. 1 and occurs under certain conditions when the frequency difference of both first-order sidebands of this FM spectrum fits the energy difference
of the $5^2$S$_{1/2}$ ground states $\tilde{F}$ = 1, $m_{\tilde{F}}$ = 0 and $\tilde{F}$ = 2, $m_{\tilde{F}}$ = 0. This resonance is used to adjust the microwave oscillator frequency to changes of ν$_{HFS}$ and to compensate a temperature dependent drift of the electronics.

An additional modulation of the microwave oscillator signal probes the first-order Zeeman splitting of HFS ground states via the superposition of the CPT resonances n = +2 and -2 or n = +3 and -3 which are indicated by the red and blue arrowed lines in Fig. 1, respectively. The differential probing of the magnetic field induced energy shifts, with one of the two CPT resonance
pairs, cancels or at least mitigates the influence of sensor temperature variations on the magnetic field measurement (Lammegger, 2008; Pollinger et al., 2018).

The instrument consists of a mixed signal electronics board and a laser unit, which are mounted in an instrument box inside the spacecraft body, as well as a sensor unit which is located outside of the satellite at the tip of a boom. Additionally, the instrument box and the sensor unit are connected with two optical fibres and two twisted pair cables to guide the light field to
and from the sensor unit and to control the sensor temperature (Pollinger et al., 2018).



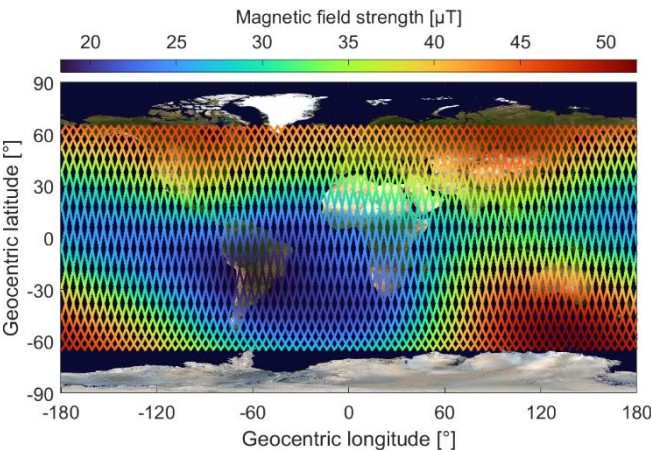

**Figure 2: Magnetic field strength measured by the CDSM between ±65° geocentric latitude within the reoccurrence period of five days. Credit for background image: Reto Stöckli, NASA Earth Observatory.**

The CDSM development started in 2007 and the instrument measured the magnetic field in space for the first time in March 2018 aboard CSES. As far as we know, this was also the first time a magnetometer based on the CPT effect was launched into space. Since then, the instrument has been operational and orbited Earth more than 9000 times until September 2019. The science phase of CSES is limited to ±65° geocentric latitude because attitude control activities aboard the satellite over the polar regions cause too much magnetic interferences. Figure 2 shows the magnetic field strength measured by the CDSM along the CSES orbit tracks between ±65° geocentric latitude for the five day reoccurrence period of 3-8 January 2019.

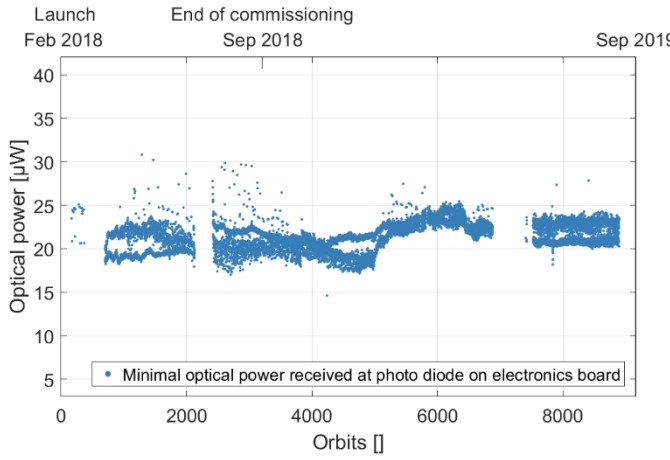

**Figure 3: Minimal optical power detected at the photo diode during each orbit.**

All available housekeeping data is within the nominal operational limits throughout the so far elapsed mission time. As an example, the minimal optical power detected at the photo diode on the electronics board is shown in Fig. 3. The light is generated in the laser unit and guided through two optical fibers to and from the sensor unit where it interacts with the rubidium atoms and derives information of the surrounding magnetic field. The optical power received at the photodiode is an indicator for the health of the VCSEL diode (Ellmeier et al., 2018), the fibers, the optical components in the sensor and the photo diode.





The graph in Fig. 3 shows gaps since not all data was made available to the CDSM team or the satellite was in a safe mode where the scientific instruments were switched off. The optical power varies due to the design of the CDSM (Pollinger et al., 2018). It is assumed that the different exposures to sunlight cause thermal stress in the multimode outbound fiber and a variation

of the polarisation state at the sensor input. After the polarizer in the sensor unit a defined linear polarisation state is re-established with the consequence that the optical power varies. No trend is visible in Fig. 3 and the minimum optical power was above the operational limit of 5 µW throughout the available data of the elapsed mission time.

Each orbit is divided into two orbit segments and data is stored separately in Hierarchical Data Format 5 (HDF5) files for each orbit segment. Dayside and nightside orbit segments are marked with the suffixes 0 and 1, respectively. For example, 44270

is the identifier for the daytime, ascending segment of orbit 4427.

## 2 Correction of in-orbit data

Several correction steps are required in order to improve the accuracy of the CDSM data. This includes the extraction of valid 1 Hz data, the application of the sensor heading characteristic, the handling of discontinuities at resonance transitions as well as the removal of fluxgate and satellite interferences. Table 1 lists these steps and introduces data product labels. L1a, L1b and

L1c are not official data products.

| Data product | Description | Section |
|:---:|:---|:---:|
| L1 | Valid 1 Hz data extracted | 2.1 |
| L1a | Sensor heading corrected | 2.2 |
| L1b | Residual discontinuity jumps at resonance transitions removed | 2.3 |
| L1c | Fluxgate feedback field cleaned | 2.4 |
| L2 | Satellite interferences cleaned – final data product | 2.5 |

**Table 1: CDSM data products and correction steps.**

### 2.1 Extraction of valid 1 Hz data

The raw data rate of the CDSM is 30 Hz. However, every second is divided in three subsequent parts: the first third of each second is reserved for the sensor heating, the second third of each second for adjusting the microwave oscillator to track the

CPT resonance $n = 0$ and the last third of each second for the actual magnetic field measurement by tracking the CPT resonance superposition $n = \pm 2$ or $n = \pm 3$.

The CPT resonances $n = 0$, $n = +2$, $n = -2$, $n = +3$ and $n = -3$ depend differently on the magnetic strength in second order (57.515 kHz mT$^{-2}$ for $n = 0$, 43.136 kHz mT$^{-2}$ for $n = \pm 2$ and 21.568 kHz mT$^{-2}$ for $n = \pm 3$). As a consequence, the CPT resonance $n = 0$ is not in the center of the CPT resonance superposition $n = \pm 2$ or $n = \pm 3$ used for magnetic field measurement.

Thus, the modulation of the microwave oscillator signal would not probe the single CPT resonances $n = +2$ and $-2$ or $n = +3$ and $-3$ at the same time. With the fact that individual single CPT resonances can have different line shapes, this might cause a





deviation of the magnetic field measurement (Pollinger et al., 2018). This cannot be ignored for CSES where the magnetic field strength is between 18-52 µT.

Therefore, during the last third of every second, the microwave oscillator control loop for tracking the CPT resonance n = 0 is
paused and the latest microwave oscillator control value is corrected by an offset in order to re-center the microwave oscillator signal with respect to the single CPT resonances n = +2 and n = -2 or n = +3 and n = -3. Details are discussed in Sect. 3.2 and (Pollinger et al., 2018).

An additional control loop tracks the CPT resonance superposition n = ±2 or n = ±3 and continuously delivers magnetic field values. However, only the last seven samples of every second are considered as unaffected by the influence of the sensor heater
current, by deviations due to the second order magnetic field dependence or by to those linked transients in the magnetic field strength read-out. These seven samples are averaged and serve as 1 Hz raw data of the CDSM instrument.

## 2.2 Application of sensor heading characteristic

The CDSM read-out has a deviation of the actual magnetic field strength which depends on the angle between the light propagation direction through the sensor and the magnetic field vector (from hereon called the sensor angle). This heading is
characteristic for the fight model and was determined during performance tests in the assembled HPM configuration at the Fragment Mountain Weak Magnetic Laboratory of the National Institute of Metrology in China (Pollinger et al., 2018). The 1 Hz raw data is corrected by this heading characteristic according to the magnetic field direction derived from the HPM fluxgate data.

As an example Fig. 4 (a) shows the magnetic field strength measured by the CDSM during orbit segment 44270. In order to
show details on the correction process, the magnetic field strength calculated with CHAOS-6 (Finlay et al., 2016; Olsen et al., 2006), a geomagnetic Earth's field model derived from Swarm, CHAMP, Ørsted and SAC-C satellite as well as ground observatory data, was subtracted.

Figure 4 (b) displays sign changed heading measurements derived from (Pollinger et al., 2018) and the angular dependent heading correction applied for the orbit segment 44270. The heading correction pattern is not continuous over an orbit segment.
The CDSM uses one of the two CPT resonance superpositions n = ±2 or n = ±3 to enable omni-directional magnetic field measurements (Pollinger et al., 2012). The selection depends on the sensor angle. For angles between approx. 0° and 60° as well as 120° to 180° the signal amplitude of the CPT resonance superposition n = ±2 is large enough to be used while for angles between approx. 60° and 120° only the superposition n = ±3 is applicable. In flight, the CDSM gets HPM fluxgate data to switch between these two resonance superpositions at 60° and 120° with an intended hysteresis of approx. 2°. The actual
switching angles vary with ±1° due to a basic on-board fluxgate correction. For the descending orbit segment 44270 shown in Fig. 4 (a) the sensor angle changed from approx. 165° to 27°. The instrument switched from the CPT resonance superposition n = ±2 to n = ±3 at approx. 117° and from the CPT resonance superposition n = ±3 to n = ±2 at approx. 58°. The heading correction for the CPT resonance superposition n = ±2 was refined compared to the linear fit in (Pollinger et al., 2018) to better represent the characteristic seen during various measurements on ground. Now, measurements with the CPT resonance



superposition n = ±2 are corrected with a second order polynomial fit. The origin of the heading characteristic is still under

investigation.

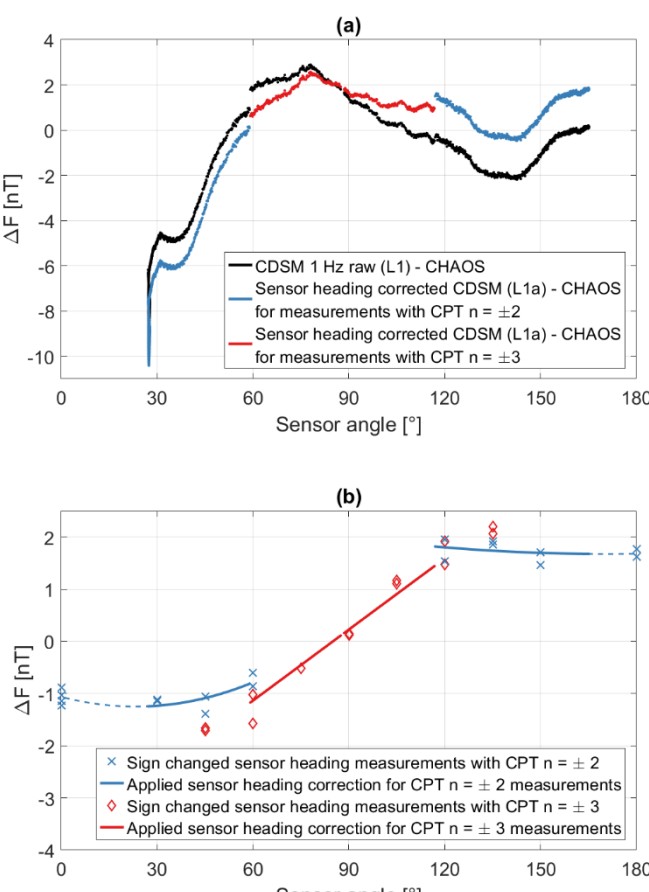

**Figure 4: Example for applying the sensor heading characteristic and the corresponding correction pattern derived from measurements on ground.**

## 2.3 Removal of residual discontinuity jumps at resonance transitions

After the sensor heading correction, the magnetic field values are not continuous when the resonance superpositions n = ±2 and n = ±3 are switched. For example, Fig. 5 (a) shows the magnetic field strength measured by the CDSM during the orbit segment 44270 with the CHAOS-6 calculation subtracted. When switching from the CPT resonance superposition n = ±2 to n = ±3 a jump of the magnetic field strength of approx. -0.71 nT is observable in the CDSM read-out while for the transition

from CPT resonance superposition n = ±3 to n = ±2 the step is approx. -0.66 nT. These discontinuity jumps vary with each orbit segment and are further discussed in Sect. 3.2.

In order to avoid a misinterpretation by the scientific user, the discontinuity jumps are removed individually for each orbit segment by adjusting the magnetic field data derived with the CPT resonance superposition n = ±3 to measurements with the CPT resonance superposition n = ±2.





At the transition from CPT resonance superposition n = ±2 to n = ±3 the signal amplitude of the CPT resonance n = 0 is also small and the microwave oscillator control loop is paused (Pollinger et al., 2018). For the subsequent measurements with the CPT resonance superposition n = ±3 the last microwave oscillator control value is re-centered as discussed in Sect 2.1. At the transition from the CPT resonance superposition n = ±3 to n = ±2 the signal of the CPT resonance n = 0 is large enough to re-activate the control loop. Consequently, for measurements with the CPT resonance superposition n = ±2 the microwave

oscillator control loop is active. Then it can compensate a possible temperature drift of the electronics and can follow a change of the HFS ground state splitting due to e.g. a sensor temperature drift.

For each orbit segment the two discontinuity jumps at the resonance transitions are used to calculate a linear ramp. The ramp is added to the magnetic field strength measured with CPT resonance superposition n = ±3. As an example this correction pattern is shown for the orbit segment 44270 in Fig. 5 (b).

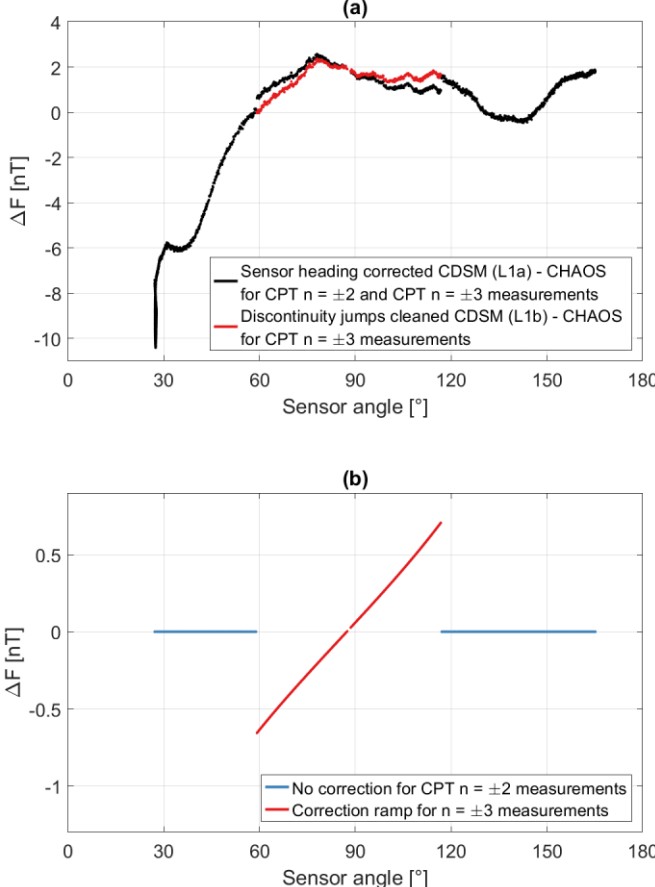


**Figure 5: Example for removing residual discontinuity jumps at the resonance transitions and the corresponding correction pattern.**





## 2.4 Removal of fluxgate interferences

The HPM sensor configuration consists of the CDSM sensor mounted at the tip of a 4.7 m long boom while the FGM 2 and

FGM 1 sensors are located 367 mm and 767 mm inwardly.

Fluxgates are inherently zero field detection devices where an artificial magnetic field is applied to cancel the environmental magnetic field in the sensor (Auster, 2008). For CSES this field can significantly influence the magnetic field measurement of the other sensors. The cross interferences were characterized with the sensors mounted on a dummy boom in a µ-metal chamber (Zhou et al., 2018). The FGM 1 and FGM 2 sensors were located at the correct distances and orientation with respect to the

CDSM position. The CDSM was replaced by a third fluxgate sensor for this test. Relevant for the CDSM is the feedback field generated by the FGM 2 sensor while the interference of FGM 1 is weak enough to be ignored. The influence of the FGM 2 feedback field $F_{FG}$ at the CDSM sensor position is

$$\begin{bmatrix} F_{FG,x_{FG}} \\ F_{FG,y_{FG}} \\ F_{FG,z_{FG}} \end{bmatrix} = I \begin{bmatrix} F_{x_{FG}} \\ F_{y_{FG}} \\ F_{z_{FG}} \end{bmatrix} = 10^{-5} \begin{bmatrix} 5.34 & 1.97 & 0.67 \\ 1.33 & -7.82 & 0.00 \\ 0.00 & 1.90 & 2.76 \end{bmatrix} \begin{bmatrix} F_{x_{FG}} \\ F_{y_{FG}} \\ F_{z_{FG}} \end{bmatrix}$$

where I is the matrix characteristic of the FGM 2 feedback field influence which depends on the magnetic Earth's field vector

F. The fluxgate coordinates $x_{FG}$, $y_{FG}$, and $z_{FG}$ correspond to the satellite coordinates $y_{sat}$, $z_{sat}$ and $x_{sat}$, respectively, where $x_{sat}$ is the flight direction and $z_{sat}$ points to the center of Earth. The CDSM scalar measurement is transformed into a vector as a function of F derived by FGM 2 and is corrected by $F_{FG}$. In orbit the impact of the FGM 2 sensor is up to 3.9 nT at the CDSM position and depends on the magnetic field direction and strength. As an example the influence of FGM 2 during orbit segment 44270 is shown in Fig. 6.

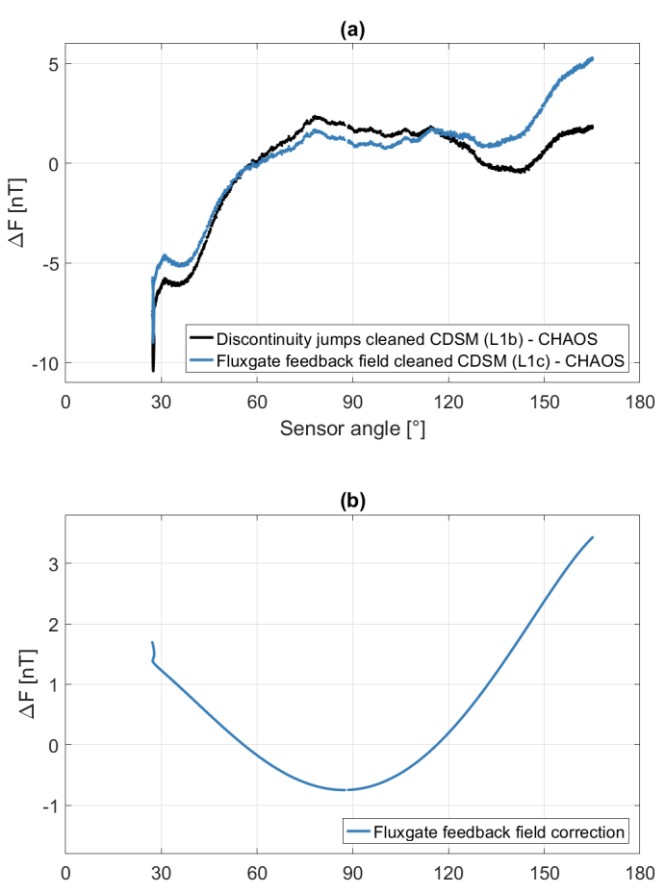


**Figure 6: Example for the influence of the FGM 2 fluxgate feedback field and the corresponding correction pattern.**

**2.5 Removal of satellite interferences**

Although there was a magnetic cleanliness program carried out by the satellite developer, some magnetic disturbances remain visible to the magnetometers. In order to be able to remove these interferences from the scientific data the whole satellite was installed in a coil system and different operation modes were magnetically measured. The influence of the satellite $F_{sat}$ at the CDSM sensor position was published in (Xiao et al., 2018) as

$$\begin{bmatrix} F_{sat,x_{sat}} \\ F_{sat,y_{sat}} \\ F_{sat,z_{sat}} \end{bmatrix} = A \begin{bmatrix} F_{x_{sat}} \\ F_{y_{sat}} \\ F_{z_{sat}} \end{bmatrix} + F_0 + C \begin{bmatrix} M_{x_{sat}} \\ M_{y_{sat}} \\ M_{z_{sat}} \end{bmatrix}$$

$$= 10^{-5} \begin{bmatrix} -1.108 & 0.025 & 0.725 \\ -0.350 & -2.808 & -1.100 \\ 1.225 & -1.158 & 4.658 \end{bmatrix} \begin{bmatrix} F_{x_{sat}} \\ F_{y_{sat}} \\ F_{z_{sat}} \end{bmatrix} + \begin{bmatrix} -0.16 \\ -0.26 \\ 0.29 \end{bmatrix} + \begin{bmatrix} 0.20 & 0.01 & 0.00 \\ 0.03 & 0.56 & -0.01 \\ -0.08 & 0.07 & -0.47 \end{bmatrix} \begin{bmatrix} M_{x_{sat}} \\ M_{y_{sat}} \\ M_{z_{sat}} \end{bmatrix}$$

where A is the matrix characteristic of the soft magnetic influences which depends on the magnetic Earth's field vector F, $F_0$

is the remanence of hard magnetic materials and C is the matrix characteristic of the magnetorquer influence which depends





on the torque states M. The coordinates $x_{sat}$ $y_{sat}$ and $z_{sat}$ correspond to the satellite where $x_{sat}$ is the flight direction and $z_{sat}$ points to the center of Earth. The CDSM scalar measurement is transformed into a vector as a function of F derived by FGM 2 and is corrected by $F_{sat}$. As an example, the influence of the satellite during the orbit segment 44270 is shown in Fig. 7.

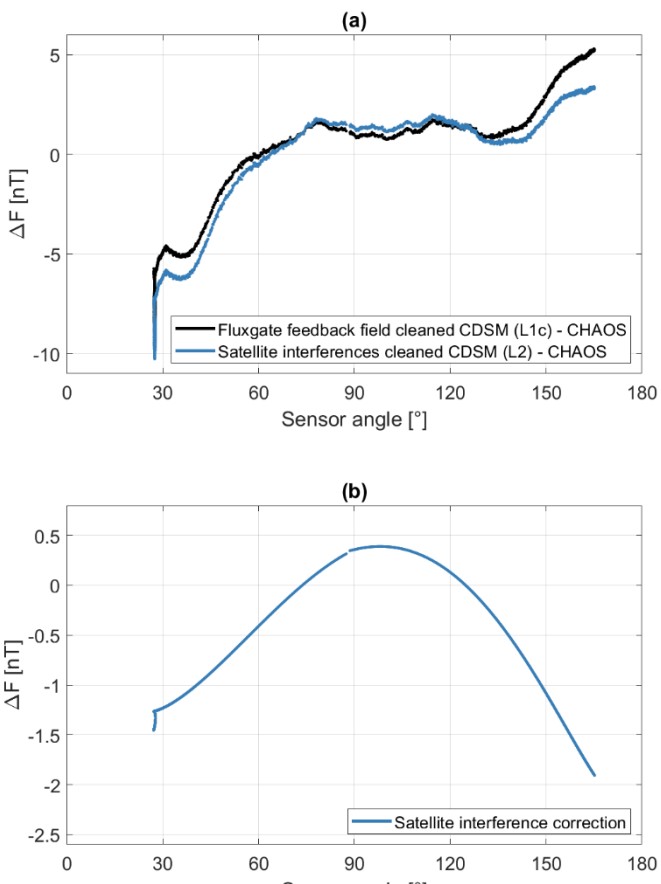

**Figure 7: Example of the satellite interferences and the corresponding correction pattern.**

## 3 In-orbit performance

The CDSM is the magnetometer with the lowest absolute error aboard CSES. Therefore, the in-orbit performance of the CDSM can solely be obtained by comparing its measurements to magnetic field models, measurements from other satellite missions or through a study of the integrity of its own data.

### 3.1 Comparison to CHAOS model and SWARM data

The CDSM data was compared to the CHAOS-6 model (Finlay et al., 2016; Olsen et al., 2006). The CHAOS model is optimized for the nightside, which means that the CHAOS coefficients are determined in such a way as to minimize the





difference between the CHAOS model and Earth's magnetic field on the nightside. These residuals are dominated by a magnetospheric ring-current distribution, which is not included in the CHAOS model and which shows a minimum scatter

around ±35° dipole latitude. Therefore, the mean values and standard deviations which are calculated for the dipole latitude ranges of -40° to -30° (southern evaluation interval) and 30° to 40° (northern evaluation interval) are an indicator for the magnetometer's data quality (Olsen, 2019). Additionally, only data with a Kp-index smaller than 1 has been selected for this evaluation. The Kp-index quantifies disturbances in the horizontal component of the Earth's magnetic field (Bartels et al., 1939).

With an ascending node at approx. 02:00 and an inclination of approx. 97°, the actual local time of CSES nightside orbit segments is between approx. 01:00 and 03:00. SWARM is a three satellite low Earth orbit mission of the European Space Agency launched in 2013 to study the Earth's magnetic field. Each satellite contains an Absolute Scalar Magnetometer (ASM) as reference instrument. The SWARM satellite Bravo has an inclination of approx. 88° and the ascending and descending nodes drift. Between 15-30 November 2018 the ascending nodes of SWARM Bravo were between 02:38 and 01:19 and 48-

42 minutes after the ascending nodes of CSES. The local time ranges overlapped for the SWARM and CSES nightside orbit segments. Data of this time interval has been selected for the comparison. The altitude of the SWARM satellite Bravo was between 501 and 518 km while CSES orbited at 500-511 km during the selected time interval.



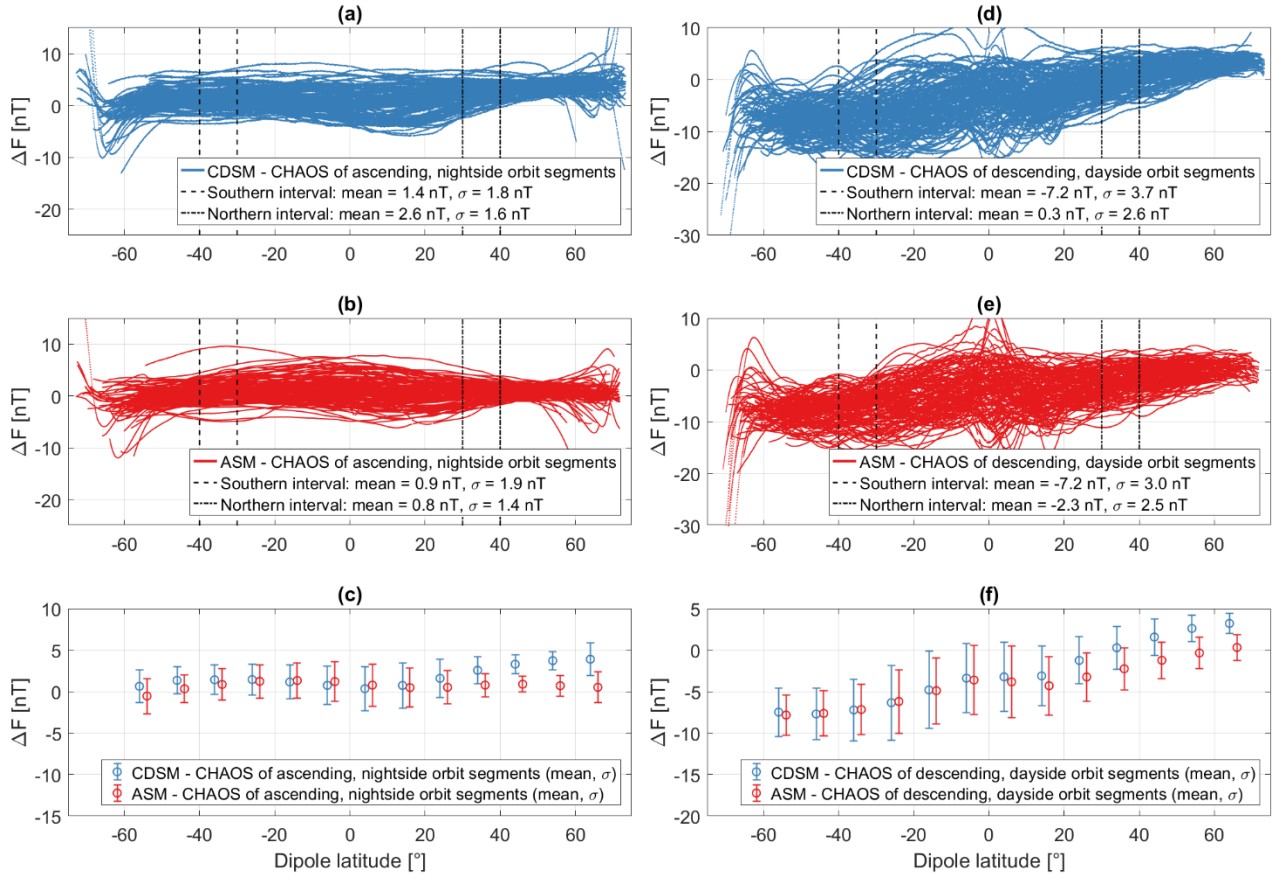

**Figure 8: Magnetic field strength measured by CDSM compared to SWARM Bravo ASM via the CHAOS-6 Earth's field model for nightside and dayside orbit segments.**

Figure 8 (a) shows the difference between CDSM measurements and the CHAOS-6 model for the 135 selected nighttime orbit segments, while Fig. 8 (b) displays the equivalent analysis for the ASM aboard SWARM satellite Bravo. The mean values $\overline{\Delta F}$ and standard deviations σ of the differences to the CHAOS model were calculated with a ten degree resolution of the dipole latitude. These values are shown as error bars for each individual instrument in Fig. 8 (c).

The mean values of both instrument deviations are consistent in the magnetic dipole latitude range of -40° to -30° ($\overline{\Delta F}$ = 1.4 nT, σ = 1.8 nT for CDSM and $\overline{\Delta F}$ = 0.9 nT, σ = 1.9 nT for ASM). One can see that the 1-σ error bars of both instruments overlap widely but start to separate at dipole latitudes greater than 20°. For the dipole latitude range of 30° to 40° the mean values of both instruments differ by 1.8 nT ($\overline{\Delta F}$ = 2.6 nT for CDSM and $\overline{\Delta F}$ = 0.8 nT for ASM). Similar differences between the CDSM and the ASM mean values can also be observed for dayside orbit segments in Fig. 8 (d), (e) and (f). These deviations from the CHAOS-6 model and the SWARM data are considered CSES specific and are still under investigation.





## 3.2 Discussion of data integrity

For the analysis in this section data from 342 orbit segments between 17-28 November 2018 and 12-18 December 2018 was available. The data set does not include orbit segments over China. As already discussed in Sect. 2.3 the magnetic field values are not continuous when the resonance superpositions $n = \pm 2$ and $n = \pm 3$ are switched. Figure 9 shows the mean, minimum

and maximum values for these discontinuity jumps for the entire data set.

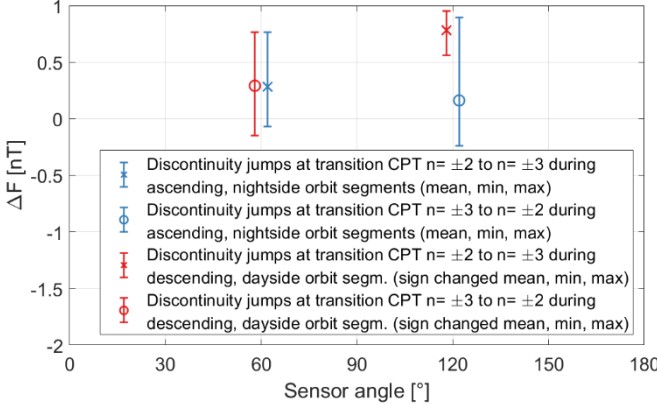

**Figure 9: Mean, minimum and maximum values of the discontinuity jumps at the resonance transitions.**

The crossed plot marks at 62° and 118° describe the change of the magnetic field strength read-out introduced by transitions from the CPT resonance superposition $n = \pm 2$ to $n = \pm 3$ for nightside and dayside orbit segments, respectively. The circled plot

marks at 122° and 58° show the discontinuity jumps at the transitions from the CPT resonance superposition $n = \pm 3$ to $n = \pm 2$ for nightside and dayside orbit segments, respectively. The sign of the values for the dayside orbit segments was changed to make them comparable to the nightside orbit segments for similar sensor angles. Ideally, the mean values should be zero but are 0.29 nT and 0.28 nT at the transitions at 58° and 62°. At the transitions at 118° and 122° the mean values (0.78 nT and 0.16 nT) differ by 0.62 nT. The maximum observed discontinuity jump is 0.95 nT which occurred at the transition from the

CPT resonance superposition $n = \pm 2$ to $n = \pm 3$ during a dayside orbit segment at 118°.

To better understand this finding all available instrument parameters and especially the microwave oscillator frequency controller adjustment were investigated in detail. The sensitivity of the magnetic field measurement as a function of a microwave oscillator frequency detuning (from hereon called detuning sensitivity) can be determined in orbit for measurements with the CPT resonance superposition $n = \pm 2$. As discussed in Sect. 2.1 every second is divided in three

subsequent parts. During the second third of each second, the microwave oscillator frequency controller tracks the HFS ground state splitting. In the last third of each second, this controller is paused and the latest control value is adjusted by an offset in order to re-center the microwave oscillator frequency with respect to the single CPT resonances $n = +2$ and $n = -2$. The CPT resonance $n = 0$ and the CPT resonance superposition $n = \pm 2$ depend differently on the magnetic field strength in second order. The applied offset is half of this frequency difference and thus a function of the magnetic field strength. The control loop for

the magnetic field measurement is active all time and read-outs can be derived during the microwave oscillator tracking and





offset parts, separately. In orbit the magnetic field strength changes with up to 40 nT s⁻¹ and, therefore, measurements done during the tracking part of each second have been interpolated to make them comparable with the offset part of each second. The impact on the magnetic field measurement as a function of the applied microwave controller offset can be used to calculate the detuning sensitivity.

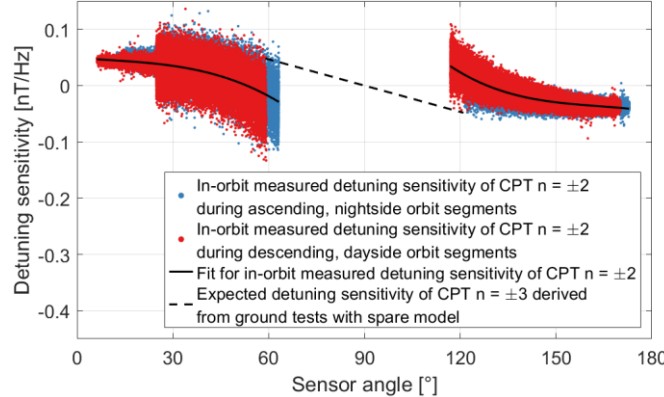


**Figure 10: Detuning sensitivity of the magnetic field measurement as a function of the microwave oscillator detuning.**

The blue and red dots in Fig. 10 show the calculated detuning sensitivity for in-orbit measurements with the CPT resonance superposition $n = \pm2$. The detuning sensitivity depends on the sensor angle. The scatter of the measured detuning sensitivity is a function of the magnetic field strength. It is dominated by the division through the microwave oscillator controller offset and

increases with decreasing magnetic field values and thus smaller offset values. This can be observed via the South Atlantic Anomaly which keeps the noise level quite high towards lower sensor angles in the southern hemisphere during many orbits. It also explains the step like drop of the noise at a sensor angle of 25°. The black solid lines are a fit of the in-orbit measurements whose shape was confirmed with the flight spare model on ground.

For the measurements with the CPT resonance superposition $n = \pm3$ the detuning sensitivity cannot be calculated from in-orbit

data. The microwave oscillator controller cannot track the HFS ground state splitting and the latest control value during measurements with the CPT resonance superposition $n = \pm2$ is always adjusted by an offset as a function of the current magnetic field strength. The detuning sensitivity for measurements with the CPT resonance superposition $n = \pm3$ shown in Fig. 10 was derived from measurements with the flight spare model.

The detuning sensitivity crosses zero at 53° and 127° for measurements with the CPT resonance superposition $n = \pm2$ and at

90° for measurements with the CPT resonance superposition $n = \pm3$. At these sensor angles the magnetic field measurement is not sensitive to the (offset) detuning of the microwave oscillator frequency with respect to the center of the single CPT resonances $n = +2$ and $n = -2$ or $n = +3$ and $n = -3$.



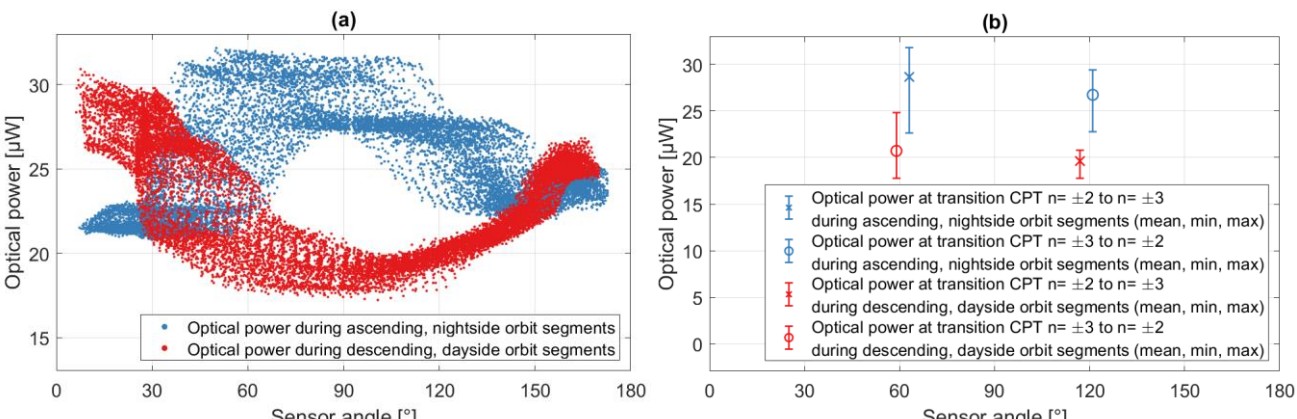

**Figure 11: Variation of the optical power.**

Figure 11 (a) displays the optical power received at the photo diode in orbit for the entire data set. It is proportional to the optical power in the sensor and varies between 17 μW and 32 μW due to the instrument design as described in the introduction. The optical power has a reoccurring pattern over time and latitude for nightside and dayside orbit segments. This is scattered when plotted as a function of the sensor angle. Figure 11 (b) shows the mean, minimum and maximum values of the optical power when the resonance superpositions n = ±2 and n = ±3 are switched.

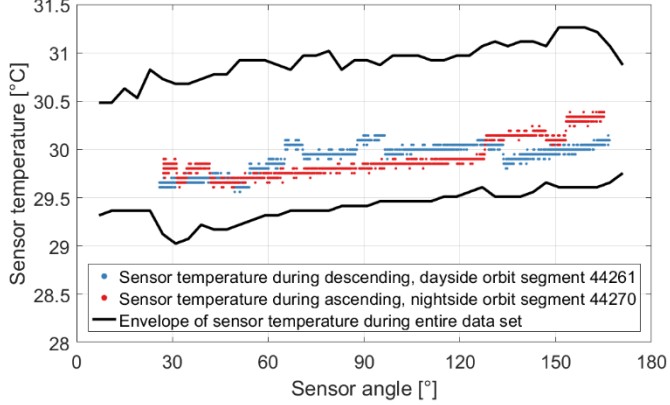


**Figure 12: Sensor temperature during orbit segments.**

The black lines in Fig. 12 show the envelope of the sensor temperature for the entire data set which is between 29.0°C and 31.3°C. The measurement experiences step-like interferences which can be observed with the sample orbit segments 44261 and 44270. These are likely caused by the unshielded twisted pair cable along the boom in combination with the high gain of 285 the measurement circuit in order to minimize the current through the platinum resistance temperature detector close to the sensor cell. For further analysis the data was filtered.





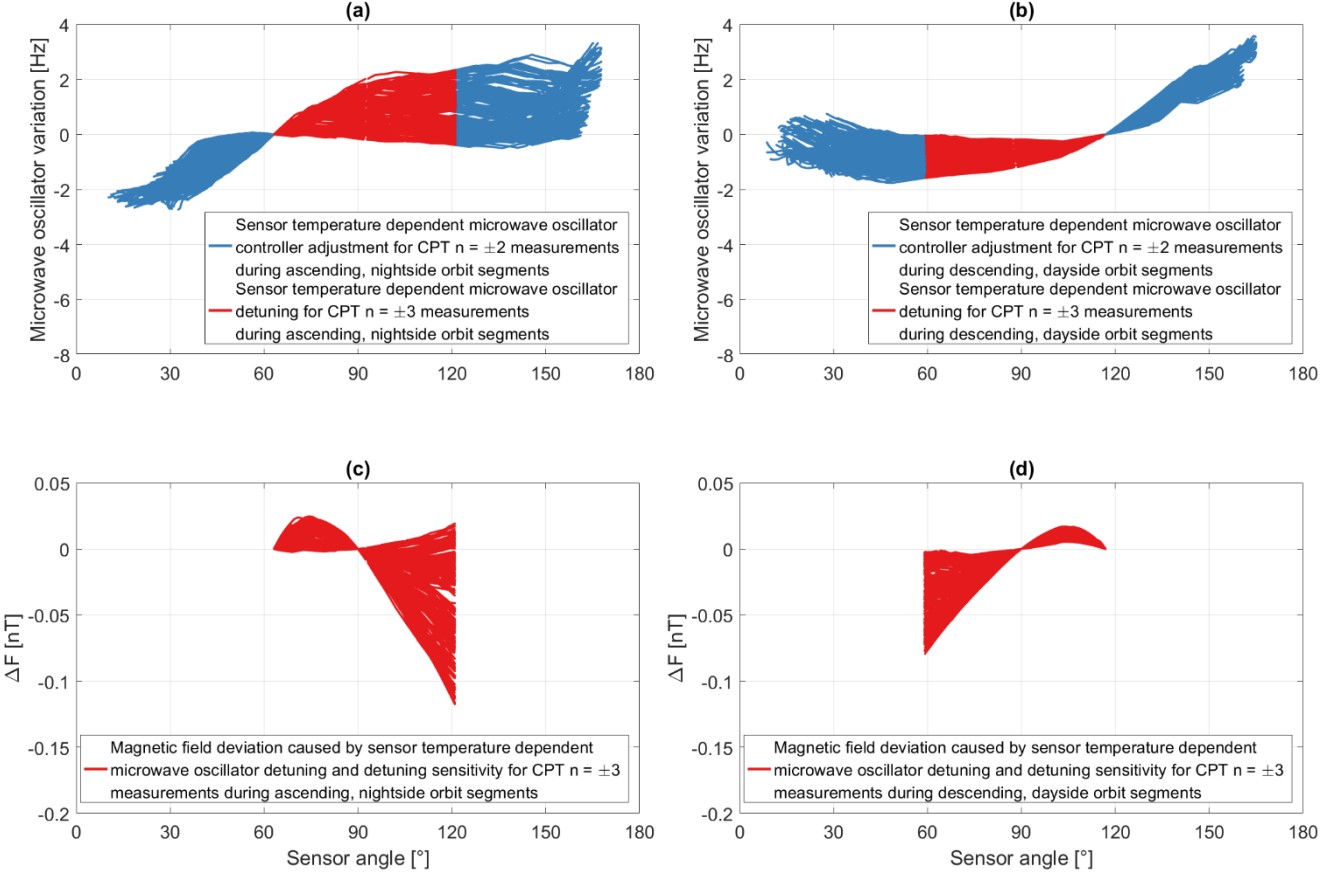

**Figure 13: Sensor temperature dependent microwave oscillator variation and magnetic field deviation.**

The HFS ground state splitting frequency depends on the sensor temperature with 13 Hz $K^{-1}$ (Pollinger et al., 2018). Figure

13 (a) and (b) show the sensor temperature dependent variations of the microwave oscillator for the entire data set. The

variations are offset with the value at the transition from CPT resonance superposition n = ±2 to n = ±3 which occurs at the

sensor angles of approx. 62° and 118° for nightside and dayside orbit segments, respectively. For measurements with the CPT

resonance superposition n = ±2 the microwave oscillator controller is active and can compensate the sensor temperature

dependent frequency changes of the HFS ground state splitting via the CPT resonance n = 0. The adjustment values are shown

as blue lines. For measurements with the CPT resonance superposition n = ±3 the microwave oscillator controller is paused

and does not track the sensor temperature dependent frequency changes of the HFS ground state splitting. A temperature

change leads to a detuning of the microwave oscillator frequency with respect to the center of the single CPT resonances n = +3

and n = -3. This detuning is shown in Fig. 13 (a) and (b) as red lines with a maximum detuning of 2.3 Hz and -1.6 Hz for

nightside and dayside orbit segments, respectively. In combination with the detuning sensitivity discussed in Fig. 10 the

detuning can cause a deviation of the magnetic field measurement with the CPT resonance superposition n = ±3. The derived

magnetic field deviation is shown in Fig. 13 (c) and (d) with a maximum deviation of the magnetic field strength of -0.11 nT





and -0.08 nT for nightside and dayside orbit segments, respectively. A sensor temperature change can contribute to the discontinuity jumps at transitions from CPT resonance superposition n = ±3 to n = ±2 in Fig. 9 but cannot affect transitions from CPT resonance superposition n = ±2 to n = ±3.

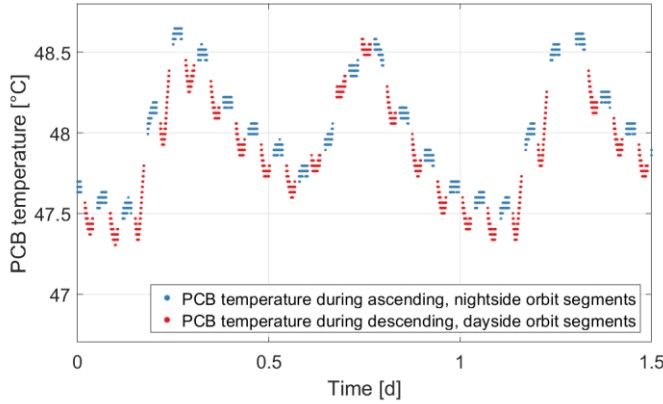

**Figure 14: Reoccurring PCB temperature pattern during orbit segments.**

The temperature of the Printed Circuit Board (PCB) is between 47.3°C and 48.8°C for the entire data set. It has a reoccurring pattern which is displayed for 1.5 days in Fig. 14. The maximum temperature change is 0.03 K per minute which occurs during specific dayside orbit segments.

The impact of the PCB temperature variations in space was investigated with the flight spare model on ground. The microwave generator is realized by a phase-locked loop which consists of a voltage-controlled microwave oscillator and a fractional n-counter frequency divider (Pollinger et al., 2018). The time base for the microwave oscillator is an adjustable reference oscillator which is tuned via a voltage input by the actuating variable of the microwave oscillator controller. The reference oscillator is temperature-compensated and autonomously adjusts the output as a function of the environmental temperature in order to mitigate the temperature dependence of the oscillator.

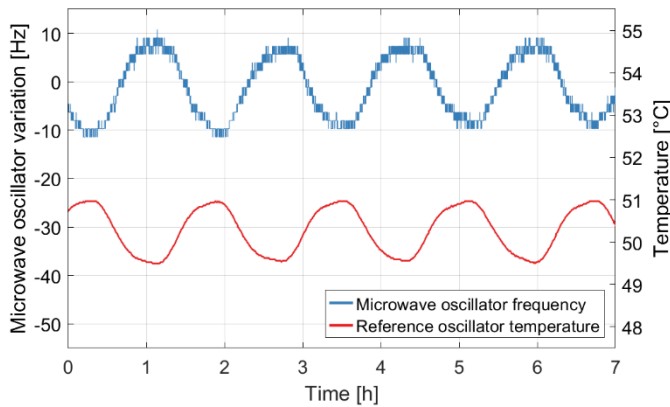

**Figure 15: Temperature dependence of the microwave oscillator output frequency.**





The temperature dependence of the reference oscillator was evaluated with the instrument box of the flight spare model, located in the thermally-controlled environment of a vacuum chamber. The output frequency was measured with a HP5335A counter
and a SRS FS725 rubidium frequency standard. The instrument box and the counter were connected via an electrical vacuum feedthrough. The reference oscillator temperature was derived from the CDSM housekeeping data since the PCB temperature measurement is within 0.5 cm on the electronics board. The reoccurring pattern of the in-orbit PCB temperature cannot be reproduced exactly with the available test facilities. Figure 15 shows the frequency change for a temperature variation of 1.4 K within an orbit period of approx. 95 minutes and a maximum temperature change of 0.07 K per minute. The reference oscillator
frequency varies which is equivalent to a change of the microwave oscillator frequency of -14.8 Hz K$^{-1}$.

The noise of the microwave oscillator control loop was evaluated with the instrument box of the flight spare model located in the thermally-controlled environment of a vacuum chamber and the sensor unit positioned outside in a μ-metal shielding can. The instrument box and the sensor unit were connected via optical and electrical vacuum feedthroughs. The maximum duration of measurements with the CPT resonance superposition n = ±3 is 13 minutes during each orbit segment for the entire data set.
This is longer than the two individual measurement intervals with the CPT resonance superposition n = ±2 during each orbit segment for the entire data set. The sensor temperature was controlled by the CDSM electronics and the CDSM housekeeping read-out varied within 0.01°C for the evaluation period of 13 minutes. The PCB temperature was kept constant by the vacuum chamber and the CDSM housekeeping read-out varied within 0.05°C for the evaluation period of 13 minutes. An artificial magnetic field was generated in the μ-metal shielding can with a Keithley 6221 current source and a coil. The generated
magnetic field strength can be assumed to be sufficiently constant for this evaluation. The microwave oscillator controller tracked the CPT resonance n = 0 and the actuating variable is a measure for the adjustment of the microwave oscillator output frequency. The standard deviation σ of the calculated microwave oscillator output frequency is 0.6 Hz for the evaluation period of 13 minutes.






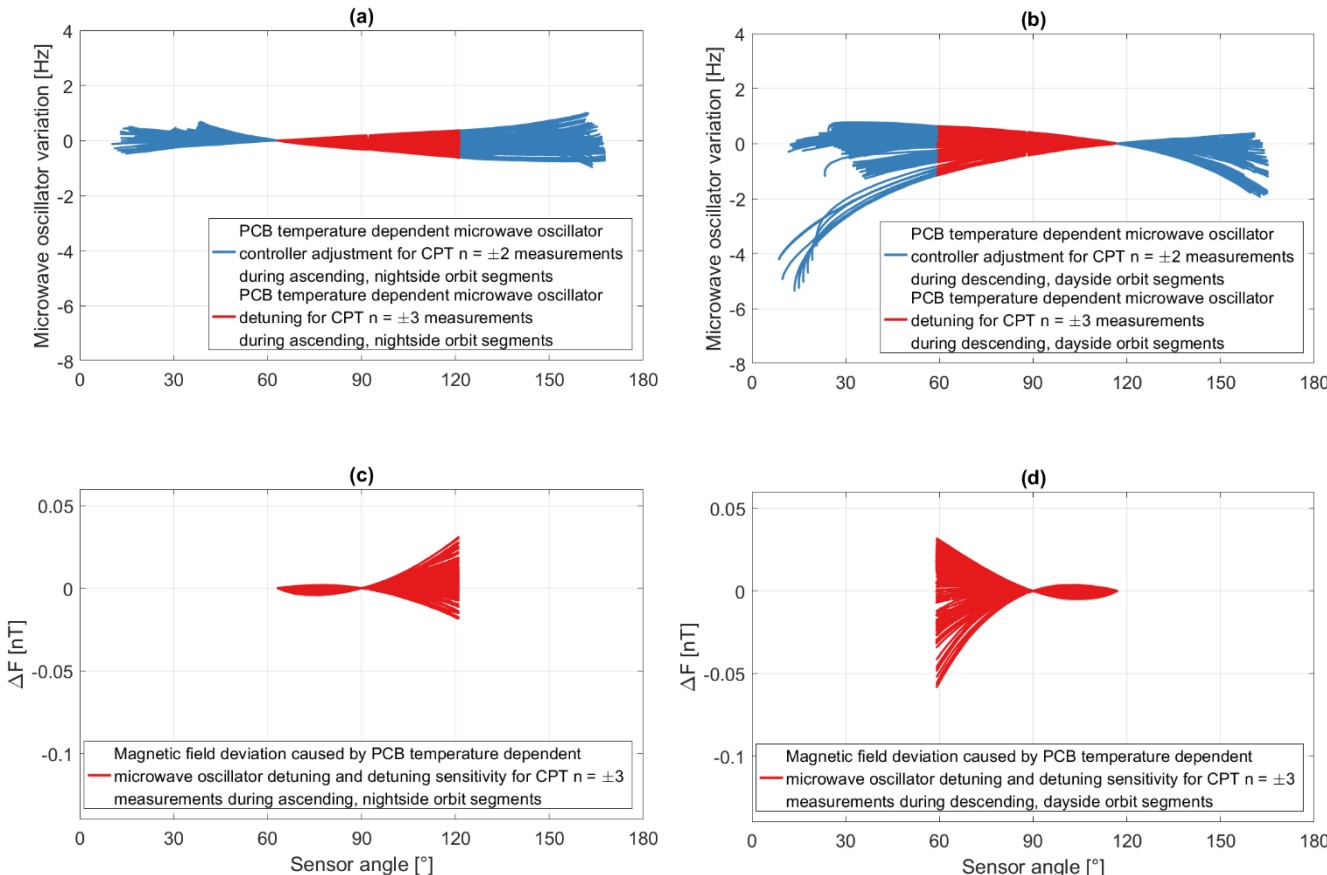

**Figure 16: PCB temperature dependent microwave oscillator variation and magnetic field deviation.**

Figure 16 (a) and (b) show PCB temperature dependent variations of the microwave oscillator for the entire data set. The analysis for the adjustment and detuning values is identical to the sensor temperature. The variations are offset with the value

at the transition from CPT resonance superposition $n = \pm2$ to $n = \pm3$. The adjustment for measurements with the CPT resonance superposition $n = \pm2$ is shown as blue lines while the detuning for measurements with the CPT resonance superposition $n = \pm3$ is displayed as red lines. The derived magnetic field deviation for measurements with the CPT resonance superposition $n = \pm3$ is shown in Fig. 16 (c) and (d). The maximum detuning of -0.6 Hz and -1.2 Hz leads with the angular dependent detuning sensitivity to a maximum deviation of the magnetic field values of 0.03 nT and -0.06 nT for nightside and dayside orbit

segments, respectively. A PCB temperature change can contribute to the discontinuity jumps at transitions from CPT resonance superposition $n = \pm3$ to $n = \pm2$ in Figure 8 but cannot affect transitions from CPT resonance superposition $n = \pm2$ to $n = \pm3$.



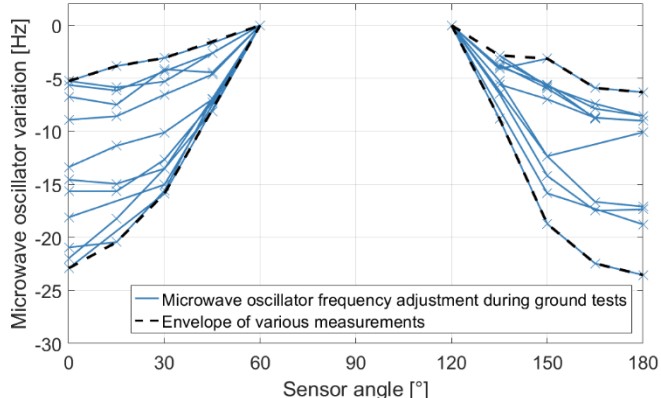

**Figure 17: Angular dependent microwave oscillator adjustment during ground tests.**

An angular dependent adjustment of the microwave oscillator frequency could be observed for measurements with the CPT
resonance superposition n = ±2. The data in Fig. 17 was derived during the sensor heading characterisation of the magnetic
field measurement with the flight model on ground. The blue lines show individual measurements at the Conrad OBServatory
(COBS) of the Zentralanstalt für Meteorologie und Geodynamik in Lower Austria and in the coil systems of the Technical
University Braunschweig (TU-BS) in Germany as well as the Fragment Mountain Weak Magnetic Laboratory of the National
Institute of Metrology (NIM) in China (see Fig. 4 (b)). The microwave oscillator variations are referenced to the sensor angles
of 60° and 120° in order to make them comparable. The black dashed lines show the envelope of these measurements. The
microwave oscillator adjustment varies between -5 Hz and -24 Hz. The sensor and PCB temperatures were settled within 0.1°C
for each run which would lead to an adjustment of the microwave oscillator frequency of only 0.7 Hz and 1.5 Hz, respectively.
The magnetic field strength was artificially controlled for the measurements at TU-BS and NIM. A magnetic field variation of
20 nT at an Earth's field of 48550 nT would lead to an adjustment of the microwave oscillator frequency of just 0.06 Hz during
the COBS measurements. Thus, the reason for the angular dependent behaviour cannot be explained so far.

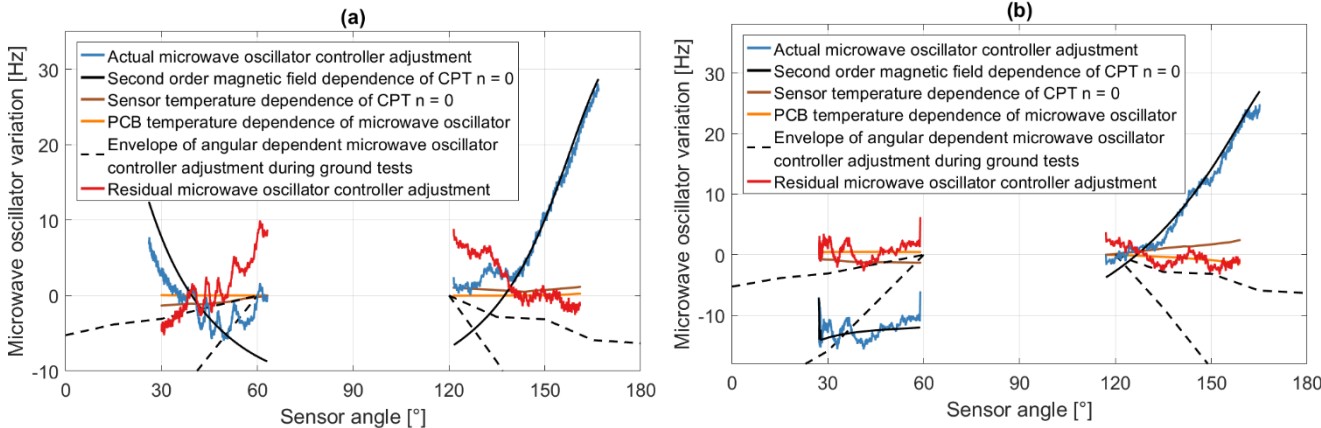

**Figure 18: Microwave oscillator variation during measurements with the CPT resonance superposition n= ±2 for individual orbit segments.**





Figure 18 shows two examples of microwave oscillator variations during measurements with the CPT resonance superposition
n = ±2 in orbit. The blue curves display the actual microwave oscillator controller adjustment required to track the CPT
resonance n = 0 with the microwave oscillator frequency. The re-centering as described in Sect. 2.1 is not displayed for
simplicity. The output frequency is offset to the last microwave oscillator controller value before it was paused at the transition
from CPT resonance superposition n = ±2 to n = ±3. For the ascending, nightside orbit segment 44261 in Fig. 18 (a) and the
descending, dayside orbit segment 44270 in Fig. 18 (b) this occurred at 62° and 118°, respectively. The brown and orange
lines are the calculated microwave oscillator adjustments needed to compensate the sensor and PCB temperature changes with
respect to the reference point at the transition from CPT resonance superposition n = ±2 to n = ±3. The black solid lines are
the expected frequency change of the CPT resonance n = 0 as a function of the magnetic field strength in second order. Their
vertical offset was obtained by nonlinear least-squares fitting to the actual microwave oscillator controller adjustment for each
individual orbit segment. The envelope of the angular dependent microwave oscillator adjustment discovered during ground
tests is shown as black dashed lines. The influences of the magnetic field strength in second order, the sensor temperature and
the PCB temperature are understood and can be subtracted from the actual microwave oscillator controller adjustment. The
residuals between the actual and understood microwave oscillator adjustments are plotted as red lines in Fig. 18 and show the
same sensor angular dependent trend as the ground measurements in Fig. 17.

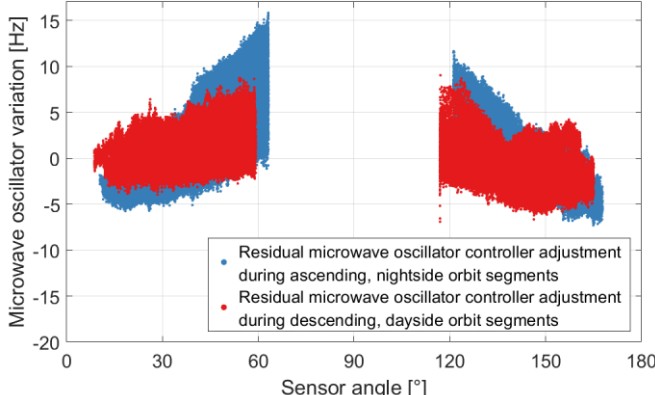

**Figure 19: Residual microwave oscillator adjustment for measurements with the CPT resonance superposition n = ±2.**

The residual microwave oscillator controller adjustments vary with each orbit segment. Figure 19 shows the residuals for the
entire data set. The maximum residual microwave oscillator adjustment is 15.8 Hz and occurs during nightside orbit segments.





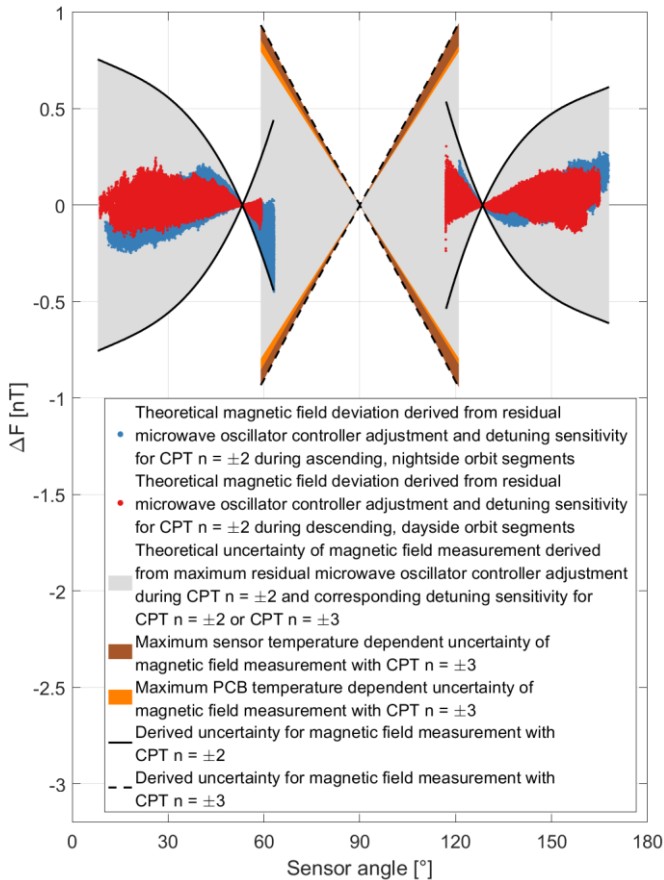

**Figure 20: Derived uncertainty of magnetic field measurement.**

For measurements with the CPT resonance superposition n = ±2 it can be assumed that the controller adjusts the microwave oscillator frequency correctly to the CPT resonance n = 0 with the limit of the control loop noise discussed above. Since the cause of the residual microwave oscillator adjustment in Fig. 19 is unknown, it cannot be assumed that the offset-adjusted light field matches the center of the single CPT resonances n = +2 and n = -2 for measurements with the CPT resonance superposition n = ±2. A theoretical magnetic field deviation associated with residual microwave oscillator adjustment can be calculated via

the detuning sensitivity for measurements with the CPT resonance superposition n = ±2 in Fig. 10. This deviation is shown as blue and red dots in Fig. 20. Taking the maximum residual microwave oscillator adjustment and the detuning sensitivity for measurements with the CPT resonance superposition n = ±2 one can derive an uncertainty for the magnetic field measurements with the CPT resonance superposition n = ±2 for the available 342 orbit segments. This conservative approach can be justified with the limited data set made available for this analysis. In Fig. 20 this uncertainty is visualized with solid back lines and grey

areas below for sensor angles between approx. 8° and 62° as well as 118° and 168°. The maximum derived uncertainty for measurements with the CPT resonance superposition n = ±2 is ±0.76 nT.





As mentioned above, for the measurements with the CPT resonance superposition n = ±3 the microwave oscillator control loop is paused at the transition from CPT resonance superposition n = ±2 to n = ±3 and the last control value is offset as a function of the current magnetic field strength. The influence of the sensor and PCB temperature changes during measurements

with the CPT resonance superposition n = ±3 could be mitigated with correction curves. Temperature dependent correction terms could be additionally applied to the last control value in order to compensate a change of the HFS ground state splitting or a temperature drift of the microwave oscillator frequency. This is implemented in the flight model but would require a regular update of certain parameters which is not applicable for this mission. The uncertainties of the magnetic field measurement caused by sensor and PCB temperature changes without correction terms are shown in Fig. 20 as brown and

orange areas, respectively. The uncertainties were defined as the absolute maximum deviations in Fig. 13 (c) and Fig. 13 (d) as well as Fig. 16 (c) and Fig. 16 (d) as a function of the sensor angle.

With the observed residual microwave oscillator adjustment during measurements with the CPT resonance superposition n = ±2 it can be assumed that similar additional adjustments would be required to re-center the light field with respect to the single CPT resonances n = +3 and n = -3 during measurements with the CPT resonance superposition n = ±3. Taking the

maximum residual microwave oscillator adjustment during measurements with the CPT resonance superposition n = ±2 and the expected detuning sensitivity for measurements with the CPT resonance superposition n = ±3 one can calculate a theoretical uncertainty associated with the expected microwave oscillator detuning during measurements with the CPT resonance superposition n = ±3. In Fig. 20 this uncertainty is visualized as grey areas for sensor angles between approx. 58° and 122°.

The sum of the sensor temperature dependent uncertainty, the PCB temperature dependent uncertainty and the uncertainty

derived from the expected microwave oscillator detuning during measurements with the CPT resonance superposition n = ±3 can be interpreted as uncertainty of the magnetic field measurement with the CPT resonance superposition n = ±3. The derived uncertainty does not exceed ±0.94 nT and is displayed in Fig. 20 with black dashed lines.

The derived uncertainties for the magnetic field measurement with the CPT resonance superposition n = ±2 or n = ±3 are in the same order of magnitude as the discontinuity jumps in Fig. 9.

## 4 Conclusion


The China Seismo-Electromagnetic Satellite (CSES) mission provides the first demonstration of the Coupled Dark State Magnetometer (CDSM) measurement principle in space. The CDSM is operational and all available housekeeping data has been nominal throughout the so far elapsed mission time.

Data correction processes were established in order to improve the accuracy of the CDSM data. This includes the extraction

of valid 1 Hz data, the application of the sensor heading characteristic, the handling of discontinuities at CPT resonance transitions as well as the removal of fluxgate and satellite interferences. The sum of all corrections applied to the CDSM L1 data is between -2.4 nT and 3.2 nT.



The CDSM measurements were compared to the Absolute Scalar Magnetometer (ASM) measurements aboard the SWARM satellite Bravo via the CHAOS-6 Earth's field model between 15-30 November 2018. In this period the ascending nodes of the SWARM satellite Bravo were between 02:38 and 01:19 and 48-42 minutes after the ascending nodes of CSES at 02:00. The local time ranges overlapped. For nightside orbit segments the mean values of both instrument deviations compared to the CHAOS-6 model were $\overline{\Delta F}$ = 1.4 nT for CDSM and $\overline{\Delta F}$ = 0.9 nT for ASM in the magnetic dipole latitude range of -40° to -30°

(southern evaluation interval). For the dipole latitude range of 30° to 40° (northern evaluation interval) the mean values of both instruments differed by 1.8 nT ($\overline{\Delta F}$ = 2.6 nT for CDSM and $\overline{\Delta F}$ = 0.8 nT for ASM). Similar differences between the CDSM and the ASM mean values were also observed for dayside orbit segments. These deviations from the CHAOS-6 model and the SWARM data are considered CSES specific and are still under investigation.

  For the available data set of 342 orbit segments, discontinuity jumps up to 0.95 nT were observed in the magnetic field strength

read-out when the CDSM switched between the CPT resonance superpositions n = ±2 and n = ±3. To better understand this finding all available instrument parameters and especially the microwave oscillator frequency controller adjustment were investigated in detail. The frequency of the microwave oscillator is used to track the HyperFine Structure (HFS) ground state splitting via the Coherent Population Trapping (CPT) resonance n = 0 and is part of the light field to track the CPT resonance superposition n= ±2 or n = ±3 for the magnetic field measurement. The sensitivity of the magnetic field measurement on a

microwave oscillator frequency detuning was calculated from in-orbit measurements with the CPT resonance superposition n = ±2. For measurements with the CPT resonance superposition n = ±3 this detuning sensitivity was determined with the flight spare model on ground.

  During measurements with the CPT resonance superposition n = ±3, the microwave oscillator control loop is paused and cannot track changes of the sensor and PCB temperatures. The maximum deviations of the magnetic field measurement caused

by sensor and PCB temperature changes during measurements with the CPT resonance superposition n = ±3 were absolute 0.11 nT and 0.06 nT, respectively, for the available data set in orbit.

  During measurements with the CPT resonance superposition n = ±2, the microwave oscillator control loop tracks changes of the HFS ground state splitting caused by variations of the magnetic field strength in second order or the sensor temperature and it compensates a possible temperature dependent drift of the electronics. These influences are understood and can be

subtracted from the actual microwave oscillator controller adjustment. A residual microwave controller adjustment up to 15.8 Hz could be observed for the available data set of 342 orbit segments. With the maximum of this residual microwave oscillator adjustment and the calculated detuning sensitivity one can derive an uncertainty of the magnetic field measurement which depends on the sensor angle between the light propagation direction through the sensor and the magnetic field vector. This conservative approach can be justified with the limited data made available for this analysis. This derived uncertainty

does not exceed ±0.76 nT for measurements with the CPT resonance superposition n = ±2 and ±0.94 nT for measurements with the CPT resonance superposition n = ±3. It is zero at sensor angles of 53°, 90° and 127°. At these angles the magnetic

field measurement is not sensitive to a moderate detuning of the microwave oscillator frequency with respect to the center of the single CPT resonances n = +2 and n = -2 or n = +3 and n = -3.

For future missions a new sensor design was developed which reduces the sensitivity of the magnetic field measurement on the microwave oscillator frequency detuning.


**Data availability**

A limited set of data was made available to the instrument developer for this analysis. The China Earthquake Administration promised to make L2 data available for the scientific community.

**Author contributions**

AP conceived the study and prepared the manuscript. WM supervised the whole project. BC and BZ carried out ground tests to characterise the fluxgate feedback coil interference, CA and ME provided expertise on the detuning sensitivity for measurements with the CPT resonance superposition n = ±3. AB, ME, CH, IJ, RL, WM and AP contributed to the instrument development. All authors have read and approved the manuscript.

**Competing interests**

The authors declare that they have no conflict of interest.

**Acknowledgements**

The authors would like to thank Nils Olsen of the Technical University of Denmark for his support and expertise on the CHAOS magnetic field model.

**Financial support**

The China Seismo-Electromagnetic Satellite mission is a project organized and mainly funded by the China National Space Administration. The work of the Space Research Institute and the Institute of Experimental Physics was co-funded by the Austrian Space Applications Programme (project no. 859716), which is managed by the Austrian Research Promotion Agency. The work of the National Space Science Center was funded by the National Key Research and Development Program of China, which is managed the Ministry of Science and Technology (project no. 2016YFB0501503).

  



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
