# Peer review of "In-orbit results of the Coupled Dark State Magnetometer aboard the China Seismo-Electromagnetic Satellite"

_Geoscientific Instrumentation, Methods and Data Systems, 2019_

## Referee Comment (RC1) · Anonymous Referee #1 · 28 Jan 2020

The paper "In-orbit results of the Coupled Dark State Magnetometer [CDSM] aboard the China Seismo-Electromagnetic Satellite", by Andreas Pollinger and others, presents the first performance results of a novel scalar magnetometer technology. It is a sequel to an earlier paper [Pollinger et al, 2018] which presented the same CDSM in its flight-ready state. The performance results presented are derived from a subset of roughly the first year's supply of data as delivered from the new China Seismo-Electromagnetic Satellite which launched in Feb, 2018. This study has taken careful looks at sources of inaccuracy, both expected and unknown. The main points of examination relate to effects of temperature on the CDSM, and on the effects of magnetic field alignment as it relates to the optical axis of the CDSM sensor.

[Figure]

The examinations appear very thorough, finding systematic errors of comparable magnitude to other, older technologies. The new CDSM instrument technology may well find new uses on spacecraft, but performance testing of the types presented here is a necessary stage of development. I would thank the authors for placing these early results into the public domain.

I have only two discussion points to raise. One has to do with temperature measurement and control, the other with future expectations for the CDSM technology.

p 15, l 286 The authors refer here to trying to minimize current in a platinum temperature sensor. Is there a reference available that includes details of the temperature sensor and its error analysis? My read of Pollinger et al (2018) does not discuss this, but that paper does refer to temperature control loops using thermistors and bifilar heating coils. A search in that paper for the work "platinum" finds no occurrences. If not could a few details be included here: nominal temperature sensor resistance, power dissipation in platinum resistor, distance to CDSM cell, estimated magnetic contamination by the temperature sensor. Possibly a general discussion of the temperature monitoring and control regimen could be added. The statement regarding minimizing current also suggests that the temperature sense current is operating continuously. Given the three way time slicing of each second would it not have been possible to make temperature measurements during the first third of each second, and disabling the temperature sense current during the remainder of the second? What temperature measurement accuracy is required to achieve suitable values for control or compensation?

p 25, l 469 "For future missions a new sensor design was developed which reduces the sensitivity of the magnetic field measurement on the microwave oscillator frequency detuning."

This closing statement leaves the reader with several questions. What is the basis for such a redesign? What would be the benefit? Are the authors satisfied with the development progress of the CDSM to date? Given the now known inaccuracy profile for

the present CDSM are large improvements likely to be possible? What are future plans for both performance testing of this existing sensor, or generally for future redesign considerations.

All further comments are with regard to minor English usage issues. Call me old-fashioned, but for me the word "data" is a plural word ["Daten" in German], and the correct English usage requires the plural verb. Most of the following are minor corrections for that usage, plus a few typographic errors.

p 3, l 67 "All available housekeeping data is within the nominal operational limits throughout the so far elapsed mission time."

This sentence is awkward. Perhaps better would be "All available housekeeping data fall within the nominal...." Also in Fig 3 and line 68 it might be better to use the expression "minimum optical power" rather than "minimal optical power" as in p 4, l 76.

p 4, l 72 "all data was made available" to "all data were made available"

l 78 "data is" to "data are"

p 5, l 110 "fight model" to "flight model" ???

l 112 "data is" to "data are"

p 11, l 202 "data with ... has been" to " data with ... have been"

l 211 "Data ... has been" to "Data ... have been"

p 13, l 227 "data ... was available" to "data ... were available"

p 15, l 286 "data was filtered" to "data were filtered"

p 20, l 355 "data ... was derived" to "data ... were derived"

p 22, l 399 "solid back lines" to "solid black lines" ???

p 23 l 427 "data has" to "data have"

---

## Referee Comment (RC2) · Anonymous Referee #2 · 17 Feb 2020

**Contents**

[Figure]

**1  General**

- The paper investigates on the in-flight performance of the Coupled Dark State Magnetometer, a payload on the China Seismo-Electromagnetic Satellite (`CSES`, launched in 2018) forming together with the `FGM` magnetometer part the `HPM` instrument package.

- The paper is a follow-on paper of the pre-flight descriptions in Pollinger et al. 2018. This update with an extended view on the in-flight behaviour of the instrument and it's quirks may well fit into the scope of the journal. It is an thorough analysis of the specific features of the CDSM instrument layout under the flight conditions, even with a limited data subset, and discusses the correctable errors as well as it tries to quantify the unexplained and uncorrectable ones.

  This may be useful to know for the community, in particular for later users of the CSES magnetic field data products – if such products get openly distributed.

**2  Comments**

- Even an extended part of the paper is presenting detailed analysis of the partly predictable variation of the satellite, i.e. orbit-dependence parameters as the temperature of the various instrument parts on the detuning effects, a conclusion, if the uncertainties and features found are limiting the `CDSM`'s success in the instrument package as an reliable reference for the scalar in-flight calibration required for the `HPM` fluxgate magnetometer usability is not mentioned.

- Page 3, figure 2 and line 63:

  Is this truncation an idiosyncratic limitation specific to the `CDSM`, driven by a specific sensitivity, or is this a limitation to the whole `HDM` instrument package, so

the `FGM` sensors as well? A hint (or a little figure of an example) describing the type of interferences may be useful. Is it a limitation caused by high gradients or caused by a specific noise from the satellite itself? Mere rotations or attitude jitters itself should not affect the readings of a scalar field experiment.

- On several places in the paper it is mentioned, that the phenomenon `is still under investigation` or similar. Are there some ideas already on the market (for example large local gradients or high frequent satellite signals)? What is already ruled out?

- Page 5, line 106: `These seven samples are averaged and serve as 1 Hz raw data of the CDSM instrument.`:

  And how is the timestamp for this fraction calculated and set? Is it not a kind of challenge to align a patchy, *spotlight*-averaged value like this with other (presumably) *continuously* sampled and presumably averaged or filtered readings?

- Line 164: I assume, all three vector magnetometers are already calibrated beforehand?

- Page 8, line 166, `while the interference of FGM 1 is weak enough to be ignored`:

  What is the threshold or criterion of being irrelevant?

- Page 10, line 186, `where xsat is the flight direction and zsat points to the centre of Earth.`

  Because the *nominal* flight direction and the *true* flight direction may differ (as a function of the attitude control system) and in contrast to the explicit description 'is the flight direction', I assume $x_{sat}$ is still a satellite fix direction. Please clarify.

- **Page 11, line 199** `These residuals are dominated by a magnetospheric ring-current distribution, which is not included in the CHAOS model and...`:

  This may be just a misleading wording, as 'distribution' presumably should point here to a second order effect. The tool, freely available to forward calculate `CHAOS` vectors as function of time and position, is very well able to supply estimates of the external field contribution and also describing the ring-current field. There is a dedicated `RC`-time-series-index file, used to parametrise the ring-current part of the external field contribution. But of course, not all possible external field contributions can be covered by such a model, there are asymmetric or imperfectly modelled, also induced constituents, local field aligned currents, ionospheric bubbles or the equatorial electrojet at low latitudes. The recipe itself, to look at 'medium' latitudes and to keep aloof of both, the equatorial and the polar region, around +- 35 degree, is good, nevertheless.

- **Page 13, line 227:** `For the analysis in this section data from 342 orbit segments between 17-28 November 2018 and 12-18 December 2018 was available.`:

  The idiosyncratic fact, that data over China were not available is already a little bit awkward, but is the covered (not even contiguous) time period limitation driven by technical or quality reasons (for example first successfully processed, lowest activity, most complete time coverage) or caused by *other* constraints? I would naively guess, a hardware supplier would have full access to the raw data frames starting somewhere during the engineering phase of a satellite mission.

- **Page 13, ff, all section 3.2:** There are a few occasions of the phrase `the entire data set`; and even it is early in the section clearly stated, that the data set in focus is a very limited subset, this should be reminded (i.e. by using `in the entire subset`).

[Figure]

- Page 13, ff, all section 3.2. The section is quite long and partly jumps from the inspection of the in-flight data (sub-)set on one hand to results using flight spare parts on ground calibrations on the other hand and back. I suggest to introduce suitable subsections guiding orientation, for example one for the sensor temperature, one for the `PCB` temperature, and so on, finally a subsection for the paragraphs which are discussing the final, partly currently uncorrectable, integrated uncertainties.

- Page 21, line 385 ff.:

  The `sensor angle` dependency, a view from inside the sensor system, is also a dependency driven by the orbit period and position of the satellite, in particular in magnetic dipole coordinates (depending also on the mounted sensor orientation relative to the local `S/C` system). Please, to illustrate if there is spatial systematic error distribution (which may affect the scientific exploitation or may be the usability for in-flight-calibration purposes), consider to add a map of the accumulated error in magnetic coordinates for some (or all) available orbits. That may give an idea of possible pitfalls for a scientific interpretation.

- All explicit references in the paper to the Swarm satellite are using the all capital word `SWARM`. But the `ESA` mission name itself is not an abbreviation, so all occasions should be changed to `Swarm`.

- I'm also curious indeed to read about the technical strategies to overcome the described inherent error sources, but I well understand that it may be to early to reveal them to the public *here and now*.

**3 Minor Remarks**

- First I agree with anonymous referee `#1` on the need for a consistent use of the word `data` and the correction of the subsequent typos in his or her comments.

- Page 4, line 75: `After the polarizer in the sensor unit...`:

  Being not a native english speaker, I found the sentence with `After` confusing, I would prefer a spatial order (like `Behind...`). What about something like `With the polarizer passed...`?

- Page 12, line 225, `These deviations from the CHAOS-6 model and the SWARM data...`:

  The Reference of `These` seems a bit unclear to me: only the latter deviations or all the ones mentioned in this paragraph?

- Page 20, figure 18:

  While blue and red give a good contrast (the colors should be a bit brighter and less saturated in combination with black in all affected figures...), the choice of orange and brown results not in an easily readable figure (at least not without the ability to zoom in – so at least not on a paper print...).

- Page 25, line 471, `Data availability`: The missing availability of the data may limiting the interest in the paper a bit – in particular as it seems not be mentioned in this paper, if the scalar readings were finally useful to inflight-calibrate the `FGM` vector magnetic field readings. If any further information about the fate of the data is available, please update.

- Page 26, in References, line 507: `Private conversation`:

  The common phrase seems to be `Private communication`.

**4   Recommendations**

- I vote for a minor revision, as some of the suggested modifications are somewhat minor (i.e. figures and colors) or just additions (figure of an example of a geographical/geomagnetic mapping of the errors) – or covered just by adding more verbosity about the data policy and the data used, and about the impact of the findings to the role of the `CDSM` in the `HPM` package.

- If there are news about the situation on the `Data availibilty` meanwhile, this entry should be updated.

---

## Author Comment (AC1) · 13 Apr 2020

**Response by the authors**

First of all, we would like to thank both reviewers for their comments and questions, which helped us to improve the manuscript significantly.

Initially, data of 342 orbit segments were made available to the CDSM team for the analysis in Sect. "3.2 Discussion of data integrity data". The discussion paper is based on this data set. However, the CDSM team unexpectedly received data of additional 7812 and 1233 orbit segments on 25 November 2019 and 5 April 2020, respectively, which can be used for the analysis in this section. We would like to ask for permission to include this new data into the analysis because we think it would highly improve the manuscript. This was also a request of reviewer 2.

This letter is divided into four sections: First, we answer the questions of the reviewers and propose changes in the manuscript (starting on page 2 and on page 4). Then, the authors would like to change a few phrases to avoid misinterpretation and improve the readability of the manuscript (page 12). Finally, the authors propose changes with updated plots in order to include the new extended data set into the manuscript (starting on page 13).

**Reviewer comments are formatted black bold**, authors' comments and unchanged phrases are formatted black normal, newly introduced phases and values are formatted blue, deleted phases and values are formatted red strikethrough.

**Response to comments of reviewer 1**

**RC1_1: p 15, l 286 The authors refer here to trying to minimize current in a platinum temperature sensor. Is there a reference available that includes details of the temperature sensor and its error analysis? My read of Pollinger et al (2018) does not discuss this, but that paper does refer to temperature control loops using thermistors and bifilar heating coils. A search in that paper for the work "platinum" finds no occurrences. If not could a few details be included here: nominal temperature sensor resistance, power dissipation in platinum resistor, distance to CDSM cell, estimated magnetic contamination by the temperature sensor. Possibly a general discussion of the temperature monitoring and control regimen could be added. The statement regarding minimizing current also suggests that the temperature sense current is operating continuously. Given the three way time slicing of each second would it not have been possible to make temperature measurements during the first third of each second, and disabling the temperature sense current during the remainder of the second? What temperature measurement accuracy is required to achieve suitable values for control or compensation?**

Author's response: The sensor temperature is measured with a PT1000 and a direct current of approx. 14µA. The low measurement current requires a high amplification at the electronics in the spacecraft and the signal-to-interference ratio is low. The relevant thermal time constants of the sensor unit are sufficiently large so that the observed interferences in the measurement signal are not relevant for the control loop.
Additionally, the controller was not activated for the operation in orbit and constant power heats the sensor unit. The same approach was used during the sensor heading characterisation of the magnetic field measurement with the flight model on ground (Pollinger et al., 2018) where the environmental temperature was settled within 0.1°C for each run. The PT1000 is not directly glued on the Rb-filled glass cell but on an aluminium shell which is glued on the glass cell. The purpose of the aluminium shell is to keep the bifilar wound heater wires and the PT1000 in distance to the Rb-filled glass cell in order to further reduce possible interferences on the magnetic field measurement. An on-off temperature measurement is definitely worth investigating for future implementations but reduces the possibility to filter the signal in the analog domain. The main focus of the design was not the accuracy of the temperature measurement but the short-term stability during measurements with the resonance superposition n=±3 where the microwave oscillator control loop cannot be active. This paper solely describes the processing and interpretation of data. An additional paragraph of the hardware implementation of the sensor temperature measurement would make this approach inconsistent. The authors would like to avoid this.

Changes proposed: P15 L283: …The controller is not active and constant power heats the sensor unit. The same approach was used during the sensor heading characterisation of the magnetic field measurement with the flight model on ground (Pollinger et al., 2018) where the environmental temperature was settled within 0.1°C for each run. The in-orbit sensor temperature measurement experiences step-like interferences which can be observed …

**RC1_2: p 25, l 469 "For future missions a new sensor design was developed which reduces the sensitivity of the magnetic field measurement on the microwave oscillator frequency detuning." This closing statement leaves the reader with several questions. What is the basis for such a redesign? What would be the benefit? Are the authors satisfied with the development progress of the CDSM to date? Given the now known inaccuracy profile for the present CDSM are large improvements likely to be possible? What are future plans for both performance testing of this existing sensor, or generally for future redesign considerations.**

Author's response: A new sensor design is the main topic of the PhD thesis of one of the co-authors. She will publish a detailed discussion on the principle and performance in the upcoming months.

Changes proposed: For future missions a new sensor design is under development was developed where the light field passes the Rb-filled glass cell twice but with opposite helicities of the circular polarization state (Ellmeier, 2019). This which reduces the sensitivity of the magnetic field measurement on the microwave oscillator frequency detuning.

Changes proposed: Include new reference: Ellmeier, M.: Evaluation of the Optical Path and the Performance of the Coupled Dark State Magnetometer, PhD thesis, Graz University of Technology, Austria, 2019.

**RC1_3: All further comments are with regard to minor English usage issues. Call me old-fashioned, but for me the word "data" is a plural word ["Daten" in German], and the correct English usage requires the plural verb.**

Author's response: The authors agree with all corrections.

**RC1_4: Most of the following are minor corrections for that usage, plus a few typographic errors.**
**p 3, l 67 "All available housekeeping data is within the nominal operational limits throughout the so far elapsed mission time." This sentence is awkward. Perhaps better would be "All available housekeeping data fall within the nominal...." Also in Fig 3 and line 68 it might be better to use the expression "minimum optical power" rather than "minimal optical power" as in p 4, l 76.**
**p 4, l 72 "all data was made available" to "all data were made available"**
**l 78 "data is" to "data are"**
**p 5, l 110 "fight model" to "flight model" ???**
**l 112 "data is" to "data are"**
**p 11, l 202 "data with ... has been" to " data with ... have been"**
**l 211 "Data ... has been" to "Data ... have been"**
**p 13, l 227 "data ... was available" to "data ... were available"**
**p 15, l 286 "data was filtered" to "data were filtered"**
**p 20, l 355 "data ... was derived" to "data ... were derived"**
**p 22, l 399 "solid back lines" to "solid black lines" ???**
**p 23 l 427 "data has" to "data have"**

Author's response: The authors agree with all corrections.

**Response to comments of reviewer 2**

**RC2_1: Even an extended part of the paper is presenting detailed analysis of the partly predictable variation of the satellite, i.e. orbit-dependence parameters as the temperature of the various instrument parts on the detuning effects, a conclusion, if the uncertainties and features found are limiting the CDSM's success in the instrument package as an reliable reference for the scalar in-flight calibration required for the HPM fluxgate magnetometer usability is not mentioned.**

Authors' comments: This paper discusses the performance of the CDSM but not of the instrument package. The suitability of the CDSM within the HPM instrument package especially for the in-flight calibration of the fluxgate magnetometers is discussed in Zhou et al., 2019.

Changes proposed: P1 L26: … and serves as the reference instrument for the measurements done by the fluxgate sensors. The suitability of the CDSM for the in-flight calibration of the fluxgate magnetometers is discussed in Zhou et al., 2019.

**RC2_2: Is this truncation an idiosyncratic limitation specific to the CDSM, driven by a specific sensitivity, or is this a limitation to the whole HDM instrument package, so the FGM sensors as well? A hint (or a little figure of an example) describing the type of interferences may be useful. Is it a limitation caused by high gradients or caused by a specific noise from the satellite itself? Mere rotations or attitude jitters itself should not affect the readings of a scalar field experiment.**

Changes proposed: P3 L61: The main scientific objective of the CSES mission is within ±65° geocentric latitude and most of the attitude control activities are moved outside this area to the polar regions (Shen et al., 2018; Zhou et al., 2019). The data transfer is separated into a 1 Hz channel for all latitudes and a channel with higher instrument update rates for the area within ±65° geocentric latitude. The 1 Hz channel is mainly used for housekeeping purposes and is not accessible for the CDSM team.

**RC2_3: On several places in the paper it is mentioned, that the phenomenon is still under investigation or similar. Are there some ideas already on the market (for example large local gradients or high frequent satellite signals)? What is already ruled out?**

Authors' comments: Unfortunately nothing worth to publish.

**RC2_4: Page 5, line 106: These seven samples are averaged and serve as 1 Hz raw data of the CDSM instrument.: And how is the timestamp for this fraction calculated and set? Is it not a kind of challenge to align a patchy, spotlight-averaged value like this with other (presumably) continuously sampled and presumably averaged or filtered readings?**

Changes proposed: P5 L105: … those linked transients in the magnetic field strength read-out. Every second the mean value of these last seven samples is tagged with the time stamp of the fourth of last seven samples. The mean values  serve as 1 Hz raw data of the CDSM instrument.

**RC2_5: Line 164: I assume, all three vector magnetometers are already calibrated beforehand?**

Author's comments: The fluxgates have been calibrated beforehand.

**RC2_6: Page 8, line 166, while the interference of FGM 1 is weak enough to be ignored: What is the threshold or criterion of being irrelevant?**

Changes proposed:

[revised manuscript text omitted]

Changes proposed: P4 L84: Table 1: Fluxgate feedback fields  cleaned

**RC2_7: Page 10, line 186, where xsat is the flight direction and zsat points to the centre of Earth. Because the nominal flight direction and the true flight direction may differ (as a function of the attitude control system) and in contrast to the explicit description 'is the flight direction', I assume x sat is still a satellite fix direction. Please clarify.**

Changes proposed: P8 L170 and P10 L186: … where $x_{sat}$ is approx. the flight direction and $z_{sat}$ points approx. to the center of Earth.

**RC2_8: Page 11, line 199 These residuals are dominated by a magnetospheric ring-current distribution, which is not included in the CHAOS model and...: This may be just a misleading wording, as 'distribution' presumably should point here to a second order effect. The tool, freely available to forward calculate CHAOS vectors as function of time and position, is very well able to supply estimates of the external field contribution and also describing the ring-current field. There is a dedicated RC-time-series-index file, used to parametrise the ring-current part of the external field contribution. But of course, not all possible external field contributions can be covered by such a model, there are asymmetric or imperfectly modelled, also induced constituents, local field aligned currents, ionospheric bubbles or the equatorial electrojet at low latitudes. The recipe itself, to look at 'medium' latitudes and to keep aloof of both, the equatorial and the polar region, around +- 35 degree, is good, nevertheless.**

Changes proposed: P11 L198: These residuals are dominated by a magnetospheric ring-current contribution , which is not included in the CHAOS model and …

**RC2_9: Page 13, line 227: For the analysis in this section data from 342 orbit segments between 17-28 November 2018 and 12-18 December 2018 was available.: The idiosyncratic fact, that data over China were not available is already a little bit awkward, but is the covered (not even contiguous) time period limitation driven by technical or quality reasons (for example first successfully processed, lowest activity, most complete time coverage) or caused by other constraints? I would naively guess, a hardware supplier would have full access to the raw data frames starting somewhere during the engineering phase of a satellite mission.**

Author's comments: Please refer to the introduction on page 1 of this letter and to the proposed changes starting on page 13 of this letter.

**RC2_10: Page 13, ff, all section 3.2: There are a few occasions of the phrase the entire data set; and even it is early in the section clearly stated, that the data set in focus is a very limited subset, this should be reminded (i.e. by using in the entire subset)**

Author's comments: The phrase "entire data set" will be changed to "entire available data set".

**RC2_11: Page 13, ff, all section 3.2. The section is quite long and partly jumps from the inspection of the in-flight data (sub-)set on one hand to results using flight spare parts on ground calibrations on the other hand and back. I suggest to introduce suitable subsections guiding orientation, for example one for the sensor temperature, one for the PCB temperature, and so on, finally a subsection for the paragraphs which are discussing the final, partly currently uncorrectable, integrated uncertainties.**

Changes proposed: Subsections included:
3.2.1 Discontinuity jumps when switching CPT resonances superpositions
3.2.2 Microwave oscillator detuning sensitivity of the magnetic field measurement
3.2.3 Optical power
3.2.4 Sensor temperature
3.2.5 PCB temperature and noise of the microwave oscillator control loop
3.2.6 Angular dependent microwave oscillator adjustment during ground tests
3.2.7 Unknown residual microwave oscillator adjustment and derived uncertainty of magnetic field measurement

Changes proposed: P13 L228: Include introduction to Sect. 3.2 before Sect. 3.2.1. The histograms are only mentioned in case the reviewers accept the discussion with the new extended data set starting on page 13 of this letter:
As already discussed in Sect. 2.3 the magnetic field values are not continuous when the resonance superpositions n = ±2 and n = ±3 are switched. Histograms of these discontinuity jumps are presented in Sect. 3.2.1. All available instrument parameters and especially the microwave oscillator frequency controller adjustment are investigated in detail. The sensitivity of the magnetic field measurement as a function of a microwave oscillator frequency detuning is derived in Sect. 3.2.2. The variations of housekeeping parameters, such as the optical power received at the photo diode as well as the sensor and Printed Circuit Board (PCB) temperature, are discussed in Sect. 3.3.3, 3.3.4 and 3.3.5, respectively. In Sect. 3.3.6, an angular dependent adjustment of the microwave oscillator frequency is presented which could be observed for measurements with the CPT resonance superposition n = ±2 during ground tests with the flight model. Some influences are understood and can be subtracted from the actual microwave oscillator controller adjustment. The unknown residual microwave oscillator adjustment is used in Sect. 3.3.7 to derive the uncertainty of the magnetic field measurement.

**RC2_12: Page 21, line 385 ff.: The sensor angle dependency, a view from inside the sensor system, is also a dependency driven by the orbit period and position of the satellite, in particular in magnetic dipole coordinates (depending also on the mounted sensor orientation relative to the local S/C system). Please, to illustrate if there is spatial systematic error distribution (which may affect the scientific exploitation or may be the usability for in-flight-calibration purposes), consider to add a map of the accumulated error in magnetic coordinates for some (or all) available orbits. That may give an idea of possible pitfalls for a scientific interpretation.**

Changes proposed: P23 L422: Add Fig. 21 with the derived uncertainty of the magnetic field measurement as a function of geomagnetic coordinates.

[Figure]

Figure 21: Derived uncertainty of the magnetic field measurement as a function of geomagnetic coordinates.

P23 L422: … and is displayed in Fig. 20 with black dashed lines. The derived uncertainty of the magnetic field measurement as a function of geomagnetic coordinates is shown in Fig. 21.

**RC2_13: All explicit references in the paper to the Swarm satellite are using the all capital word SWARM. But the ESA mission name itself is not an abbreviation, so all occasions should be changed to Swarm.**

Author's comments: The authors agree with the correction.

**RC2_14: I'm also curious indeed to read about the technical strategies to overcome the described inherent error sources, but I well understand that it may be too early to reveal them to the public here and now.**

Author's response: Please refer to RC1_2.

**RC2_15: First I agree with anonymous referee #1 on the need for a consistent use of the word *data* and the correction of the subsequent typos in his or her comments.**

Author's comments: The authors agree with all corrections.

**RC2_16: Page 4, line 75:** *After the polarizer in the sensor unit...*: **Being not a native english speaker, I found the sentence with** *After* **confusing, I would prefer a spatial order (like Behind...). What about something like** *With the polarizer passed...***?**

Changes proposed: P4 L75: Behind  the polarizer in the sensor unit a defined linear polarisation state …

**RC2_17: age 12, line 225,** *These deviations from the CHAOS-6 model and the SWARM data...*: **The Reference of These seems a bit unclear to me: only the latter deviations or all the ones mentioned in this paragraph?**

Authors' comments: After further discussions, it's unclear where the deviations from the CHAOS-6 model and the deviations from the SWARM data come from.

Changes proposed: P12 L224 and P24 L442: Delete sentence:

**RC2_18: Page 20, figure 18: While blue and red give a good contrast (the colors should be a bit brighter and less saturated in combination with black in all affected figures...), the choice of orange and brown results not in an easily readable figure (at least not without the ability to zoom in – so at least not on a paper print...)**

Author's comments: The color selection follows the journal's recommendation of using ColorBrewer 2.0:
https://www.geoscientific-instrumentation-methods-and-data-systems.net/for_authors/manuscript_preparation.html
The authors will discuss with the journal editors at Copernicus how to improve this.

**RC2_19: Page 25, line 471, Data availability: The missing availability of the data may limiting the interest in the paper a bit – in particular as it seems not be mentioned in this paper, if the scalar readings were finally useful to inflight-calibrate the FGM vector magnetic field readings. If any further information about the fate of the data is available, please update.**

Author's comments: Please refer to the introduction on page 1 of this letter, to the proposed changes starting on page 13 of this letter and RC2_1.

**RC2_20: Page 26, in References, line 507:** *Private conversation:* **The common phrase seems to be** *Private communication.*

Author's comments: The authors agree with the correction.

**Request of authors to improve wording**

The authors would like to change the expression "resonance transition" to various forms of "switching" in order to avoid a misinterpretation with the term "resonance transition" used in optical spectroscopy:

- P1 L14, P4 L82, P23 L429: This includes the extraction of valid 1 Hz data, the application of the sensor heading characteristic, the handling of discontinuities , which occur when switching between the CPT resonances superpositions, at CPT resonance transitions as well as the removal of fluxgate and satellite interferences.
- P4 L86 Table 1: Residual discontinuity jumps when switching between the CPT resonance superpositions at resonance transitions removed
- P6 L135: Section 2.3 Removal of residual discontinuity jumps when switching CPT resonances superpositions at resonance transitions
- P6 139: … while for the change transition from CPT resonance superposition …
- P7 L145, P7 L147, P21 L372, P21 L376, P23 L403: When switching At the transition from CPT resonance superposition …
- P7 L156: Caption of Fig 5: Example for removing residual discontinuity jumps which occur when switching CPT resonances superpositions at resonance transitions and the corresponding correction pattern.
- P16 L290 and P19 L344: The variations are offset with respect to the reference points when switching with the value at the transition from CPT resonance superposition n = ±2 to n = ±3
- P17 L302 and P19 L350: … can contribute to the discontinuity jumps when switching at transitions from CPT resonance superposition n = ±3 to n = ±2 … but cannot affect the discontinuity jumps when switching transitions from CPT resonance superposition n = ±2 to n = ±3.

The authors would like to change following phrases in order to improve the manuscript:

- P1 L26: The CDSM measures the magnetic field strength scalar field with the lowest absolute error …
- P4 L74: It is assumed that the different exposures to sunlight cause thermal stress in the multimode outbound fiber which results in and a variation of the polarisation state at the sensor input.
- P8 L169 and P9 L184: … which depends on the Earth's magnetic Earth's field vector F …
- P12 L221: One can see that the 1-σ error bars of both instruments match in size and mean values overlap widely but start …
- P18 L330: This is longer than each of the two individual measurement intervals with the CPT resonance superposition n = ±2 during each orbit segment
- P24 L445: To better understand this finding all All available instrument parameters …

**Changes proposed to include new extended data set**

The proposed changes are organized according to the structure of the discussion paper and additionally to the new subsections introduced after a request of reviewer 2 (see RC2_11). **Sections and subsections are formatted black bold**, authors' comments and unchanged phrases are formatted black normal, newly introduced phases and values are formatted blue, deleted phases and values are formatted red strikethrough.

**1 Introduction**

P3 L61: Since then, the instrument has been operational and orbited Earth more than 12000  times until April 2020 .

P3 L65: Update of Fig. 3 and include maximum optical power detected at the photo diode during individual orbit segments.

[Figure]

**Figure 3: Minimum and maximum  optical power detected at the photo diode during individual  orbit segments.**

P3 L67: As an example, minimum and maximum  optical power detected at the photo diode on the electronics board is shown for individual orbit segments in Fig. 3.

**3.2 Discussion of data integrity**

P13 L227: For the analysis in this section data from 9387  of possible 13058 orbit segments between 16 November 2018 and 19 January 2020  were available.

**3.2.1 Discontinuity jumps when switching CPT resonances superpositions**

P13 L231: Update of Fig. 9 from  to histograms:

[Figure]

**Figure 9: Histograms of the discontinuity jumps when switching the CPT resonance superpositions. **

P13 L229: Update discussion of Fig. 9:

Figure 9 shows the histograms for the discontinuity jumps for the entire available data set. The blue histogram in Fig. 9 (a) describes the changes of the magnetic field strength read-out introduced by switching from the CPT resonance superposition n = ±2 to n = ±3 at sensor angles of approx. 62° during nightside orbit segments. The blue histogram in Fig. 9 (b) shows the discontinuity jumps when switching from the CPT resonance superposition n = ±3 to n = ±2 for nightside orbit segments which occur at sensor angles of approx. 122°. The red histogram in Fig. 9 (b) describes the discontinuity jumps when switching from the CPT resonance superposition n = ±2 to n = ±3 at sensor angles of approx. 118° during dayside orbit segments. The red histogram in Fig. 9 (a) shows the discontinuity jumps when switching from the CPT resonance superposition n = ±3 to n = ±2 for dayside orbit segments which occur at sensor angles of approx. 58°. The sign of the values for the dayside orbit segments was changed to make them comparable to the nightside orbit segments for similar sensor angles. Ideally, each of the four medians should be zero. There is no significant difference between the medians of 0.34 nT and 0.23 nT when switching CPT resonances superpositions at approx. 58° and 62°, respectively. However, a significant difference exists when switching CPT resonances superpositions at approx. 118° and 122° (0.72 nT and 0.15 nT, respectively). ~~Figure 9 shows the mean, minimum and maximum values for these discontinuity jumps for the entire data set. The crossed plot marks at 62° and 118° describe the change of the magnetic field strength read-out introduced by transitions from the CPT resonance superposition n = ±2 to n = ±3 for nightside and dayside orbit segments, respectively. The circled plot marks at 122° and 58° show the discontinuity jumps at the transitions from the CPT resonance superposition n = ±3 to n = ±2 for nightside and dayside orbit segments, respectively. The sign of the values for the dayside orbit segments was changed to make them comparable to the nightside orbit segments for similar sensor~~

~~angles. Ideally, the mean values should be zero but are 0.29 nT and 0.28 nT at the transitions at 58° and 62°. At the transitions at 118° and 122° the mean values (0.78 nT and 0.16 nT) differ by 0.62 nT. The maximum observed discontinuity jump is 0.95 nT which occurred at the transition from the CPT resonance superposition n = ±2 to n = ±3 during a dayside orbit segment at 118°. To better understand this finding all available instrument parameters and especially the microwave oscillator frequency controller adjustment were investigated in detail.~~

**3.2.2 Microwave oscillator detuning sensitivity of the magnetic field measurement**

P14 L255: Update of Fig. 10:

[Figure]

**Figure 10: Detuning sensitivity of the magnetic field measurement** .

P14 L258: Apart from data artefacts, the  scatter of the measured detuning sensitivity is a function of the magnetic field strength.

**3.2.3 Optical power**

P15 L274: Update of Fig. 11: The authors propose to divide Fig. 11 into two separate figures. In order to keep the discussion here clear the figures are labeled Fig. 11_1 and Fig. 11_2. For the final paper the numbering will be Fig. 11 and Fig. 12 and subsequent figures will be renumbered as well. Figure 11_1 replaces Fig. 11 (a): With 9387 instead of 342 available orbit segments the  in one plot would be confusing: Therefore, two sample orbit segments and the envelope are now shown. Figure 11_2 (a) and (b) replace Fig 11 (b) and are updated from  to histograms:

[Figure]

**Figure 11_1: Optical power during orbit segments.**

[Figure]

**Figure 11_2: Histograms of the optical power when switching the CPT resonance superpositions.**

P15 L275: Update discussion of Fig. 11:
The black lines in Fig. 11_1 show the envelope of the optical power received at the photo diode for the entire available data set. It is proportional to the optical power in the sensor and varies between 17 µW and 36 µW due to the instrument design as described in the introduction. A major part of this variation occurs every orbit, which can be observed with the sample orbit segments 44261 and 44270. For completeness, Figure 11_2 shows the histograms of the optical power when the CDSM switches between the resonance superpositions n = ±2 and n = ±3. As an example and similar to Fig. 9, the blue histogram in Fig. 11_2 (a) describes the optical power when switching from the CPT resonance superposition n = ±2 to n = ±3 at sensor angles of approx. 62° during nightside orbit segments.

time and latitude for nightside and dayside orbit segments. This is scattered when plotted as a function of the sensor angle. Figure 11 (b) shows the mean, minimum and maximum values of the optical power when the resonance superpositions n = ±2 and n = ±3 are switched.

**3.2.4 Sensor temperature**

P15 L280: Update of Fig. 12:

[Figure]

**Figure 12: Sensor temperature during orbit segments.**

P15 L282: The black lines in Fig. 12 show the envelope of the sensor temperature for the entire available data set . The major part of the variation between 26.2°C and 32.7°C is seasonal.

P16 L288: Update of Fig. 13:

[Figure]

**Figure 13: Sensor temperature dependent microwave oscillator variation and magnetic field deviation.**

P16 L298: This detuning is shown in Fig. 13 (a) and (b) as red lines with a maximum detuning of 3.3 Hz 2.3 Hz and -2.1 Hz -1.6 Hz for nightside and dayside orbit segments, respectively.

P16 L301: … maximum deviation of the magnetic field strength of -0.16 nT -0.11 nT and -0.10 nT -0.08 nT for nightside and dayside orbit segments, respectively.

**3.2.5 PCB temperature and noise of the microwave oscillator control loop**

P17 L307: The temperature of the Printed Circuit Board (PCB) is between 46.2°C 47.3°C and 49.7°C 48.8°C for the entire available data set.

P19 L342: Update of Fig. 16:

[Figure]

**Figure 16: PCB temperature dependent microwave oscillator variation and magnetic field deviation.**

P19 L348: The maximum detuning of 1.0 Hz -0.6 Hz and -2.1 Hz -1.2 Hz leads with the angular dependent detuning sensitivity to a maximum deviation of the magnetic field values of -0.05 nT 0.03 nT and -0.10 nT -0.06nT for nightside and dayside orbit segments, respectively.

**3.2.7 Unknown residual microwave oscillator adjustment and derived uncertainty of magnetic field measurement**

P21 L385: Update of Fig. 19:

[Figure]

**Figure 19: Residual microwave oscillator adjustment for measurements with the CPT resonance superposition n = ±2.**

P21 L387: The maximum residual microwave oscillator adjustment is 17.3 Hz 15.8 Hz and occurs during nightside orbit segments.

P22 L389: Update of Fig. 20

[Figure]

**Figure 20: Derived uncertainty of magnetic field measurement as a function of the sensor angle.**

P22 L397: … one can derive an uncertainty for the magnetic field measurements with the CPT resonance superposition n = ±2 for the available 9387  orbit segments.  In Fig. 20 this uncertainty is visualized with solid black lines and grey areas below for sensor angles between approx. 6°  and 62° as well as 118° and 169° . The maximum derived uncertainty for measurements with the CPT resonance superposition n = ±2 is ±0.8 nT .

P23 L421: The derived uncertainty does not exceed ±1.1 nT  and is displayed in Fig. 20 with black dashed lines.

P23 L423: With the results of Fig. 20 one can calculate the sum the derived uncertainties for the magnetic field measurement with the CPT resonance superposition n = ±2 and n = ±3 for sensor angles when the CPT resonance superpositions n = ±2 and n = ±3 are switched (see Sect. 3.2.1). The combined uncertainties for the magnetic field measurement are approx. ±1.4 nT, ±1.4 nT, ±1.5 nT and ±1.4 nT at the sensor angles of approx. 58°, 62°, 118° and 122°, respectively. 99.3%, 99.3%, 99.5% and 99.8% of the corresponding discontinuity jumps are within these combined derived uncertainties for the magnetic field measurement with the CPT resonance superposition n = ±2 and n = ±3.

**Conclusion**

P24 L444: For the available data set of 9387  orbit segments, discontinuity jumps with a median up to 0.72 nT  were observed in the magnetic field strength read-out when the CDSM switched between the CPT resonance superpositions n = ±2 and n = ±3.

P24 L455: … changes during measurements with the CPT resonance superposition n = ±3 were absolute 0.16 nT  and 0.10 nT , respectively, for the available data set in orbit.

P24 L460: A residual microwave controller adjustment up to 17.3 Hz  could be observed for the available data set of 9387  orbit segments.

P24 L461: With the maximum of this residual microwave oscillator adjustment and the calculated detuning sensitivity one can derive an uncertainty of the magnetic field measurement which depends on the sensor angle between the light propagation direction through the sensor and the magnetic field vector.  This derived uncertainty does not exceed ±0.8 nT  for measurements with the CPT resonance superposition n = ±2 and ±1.1 nT  for measurements with the CPT resonance superposition n = ±3.

---

## Author Response (AR1)

**Response by the authors**

First of all, we would like to thank both reviewers for their comments and questions, which helped us to improve the manuscript significantly.

Initially, data of 342 orbit segments were made available to the CDSM team for the analysis in Sect. "3.2 Discussion of data integrity data". The discussion paper is based on this data set. However, the CDSM team unexpectedly received data of additional 7812 and 1233 orbit segments on 25 November 2019 and 5 April 2020, respectively, which can be used for the analysis in this section. We would like to ask for permission to include this new data into the analysis because we think it would highly improve the manuscript. This was also a request of reviewer 2.

This letter is divided into four sections: First, we answer the questions of the reviewers and propose changes in the manuscript (starting on page 2 and on page 4). Then, the authors would like to change a few phrases to avoid misinterpretation and improve the readability of the manuscript (page 12). Finally, the authors propose changes with updated plots in order to include the new extended data set into the manuscript (starting on page 13).

**Reviewer comments are formatted black bold**, authors' comments and unchanged phrases are formatted black normal, newly introduced phases and values are formatted blue, deleted phases and values are formatted red strikethrough.

**Response to comments of reviewer 1**

RC1\_1: p 15, l 286 The authors refer here to trying to minimize current in a platinum temperature sensor. Is there a reference available that includes details of the temperature sensor and its error analysis? My read of Pollinger et al (2018) does not discuss this, but that paper does refer to temperature control loops using thermistors and bifilar heating coils. A search in that paper for the work "platinum" finds no occurrences. If not could a few details be included here: nominal temperature sensor resistance, power dissipation in platinum resistor, distance to CDSM cell, estimated magnetic contamination by the temperature sensor. Possibly a general discussion of the temperature monitoring and control regimen could be added. The statement regarding minimizing current also suggests that the temperature sense current is operating continuously. Given the three way time slicing of each second would it not have been possible to make temperature measurements during the first third of each second, and disabling the temperature sense current during the remainder of the second? What temperature measurement accuracy is required to achieve suitable values for control or compensation?

Author's response: The sensor temperature is measured with a PT1000 and a direct current of approx.  $14\mu$ A. The low measurement current requires a high amplification at the electronics in the spacecraft and the signal-to-interference ratio is low. The relevant thermal time constants of the sensor unit are sufficiently large so that the observed interferences in the measurement signal are not relevant for the control loop.

Additionally, the controller was not activated for the operation in orbit and constant power heats the sensor unit. The same approach was used during the sensor heading characterisation of the magnetic field measurement with the flight model on ground (Pollinger et al., 2018) where the environmental temperature was settled within 0.1°C for each run. The PT1000 is not directly glued on the Rb-filled glass cell but on an aluminium shell which is glued on the glass cell. The purpose of the aluminium shell is to keep the bifilar wound heater wires and the PT1000 in distance to the Rb-filled glass cell in order to further reduce possible interferences on the magnetic field measurement. An on-off temperature measurement is definitely worth investigating for future implementations but reduces the possibility to filter the signal in the analog domain. The main focus of the design was not the accuracy of the temperature measurement but the short-term stability during measurements with the resonance superposition n=±3 where the microwave oscillator control loop cannot be active. This paper solely describes the processing and interpretation of data. An additional paragraph of the hardware implementation of the sensor temperature measurement would make this approach inconsistent. The authors would like to avoid this.

Changes proposed: P15 L283: ...The controller is not active and constant power heats the sensor unit. The same approach was used during the sensor heading characterisation of the magnetic field measurement with the flight model on ground (Pollinger et al., 2018) where the environmental temperature was settled within 0.1°C for each run. The in-orbit sensor temperature measurement experiences step-like interferences which can be observed ... RC1\_2: p 25, I 469 "For future missions a new sensor design was developed which reduces the sensitivity of the magnetic field measurement on the microwave oscillator frequency detuning." This closing statement leaves the reader with several questions. What is the basis for such a redesign? What would be the benefit? Are the authors satisfied with the development progress of the CDSM to date? Given the now known inaccuracy profile for the present CDSM are large improvements likely to be possible? What are future plans for both performance testing of this existing sensor, or generally for future redesign considerations.

Author's response: A new sensor design is the main topic of the PhD thesis of one of the co-authors. She will publish a detailed discussion on the principle and performance in the upcoming months.

Changes proposed: For future missions a new sensor design is under development was developed where the light field passes the Rb-filled glass cell twice but with opposite helicities of the circular polarization state (Ellmeier, 2019). This which reduces the sensitivity of the magnetic field measurement on the microwave oscillator frequency detuning.

Changes proposed: Include new reference: Ellmeier, M.: Evaluation of the Optical Path and the Performance of the Coupled Dark State Magnetometer, PhD thesis, Graz University of Technology, Austria, 2019.

RC1\_3: All further comments are with regard to minor English usage issues. Call me old-fashioned, but for me the word "data" is a plural word ["Daten" in German], and the correct English usage requires the plural verb.

Author's response: The authors agree with all corrections.

RC1\_4: Most of the following are minor corrections for that usage, plus a few typographic errors. p 3, I 67 "All available housekeeping data is within the nominal operational limits throughout the so far elapsed mission time." This sentence is awkward. Perhaps better would be "All available housekeeping data fall within the nominal...." Also in Fig 3 and line 68 it might be better to use the expression "minimum optical power" rather than "minimal optical power" as in p 4, I 76. p 4, I 72 "all data was made available" to "all data were made available" I 78 "data is" to "data are" p 5, I 110 "fight model" to "flight model" ??? I 112 "data is" to "data are" p 11, I 202 "data with ... has been" to " data with ... have been" I 211 "Data ... has been" to "Data ... have been" p 13, I 227 "data ... was available" to "data ... were available" p 15, I 286 "data was filtered" to "data were filtered" p 20, I 355 "data ... was derived" to "data ... were derived" p 22, I 399 "solid back lines" to "solid black lines" ??? p 23 l 427 "data has" to "data have"

Author's response: The authors agree with all corrections.

**Response to comments of reviewer 2**

RC2\_1: Even an extended part of the paper is presenting detailed analysis of the partly predictable variation of the satellite, i.e. orbit-dependence parameters as the temperature of the various instrument parts on the detuning effects, a conclusion, if the uncertainties and features found are limiting the CDSM's success in the instrument package as an reliable reference for the scalar in-flight calibration required for the HPM fluxgate magnetometer usability is not mentioned.

Authors' comments: This paper discusses the performance of the CDSM but not of the instrument package. The suitability of the CDSM within the HPM instrument package especially for the in-flight calibration of the fluxgate magnetometers is discussed in Zhou et al., 2019.

Changes proposed: P1 L26: ... and serves as the reference instrument for the measurements done by the fluxgate sensors. The suitability of the CDSM for the in-flight calibration of the fluxgate magnetometers is discussed in Zhou et al., 2019.

RC2\_2: Is this truncation an idiosyncratic limitation specific to the CDSM, driven by a specific sensitivity, or is this a limitation to the whole HDM instrument package, so the FGM sensors as well? A hint (or a little figure of an example) describing the type of interferences may be useful. Is it a limitation caused by high gradients or caused by a specific noise from the satellite itself? Mere rotations or attitude jitters itself should not affect the readings of a scalar field experiment.

Changes proposed: P3 L61: The main scientific objective of the CSES mission is within  $\pm 65^{\circ}$  geocentric latitude and most of the attitude control activities are moved outside this area to the polar regions (Shen et al., 2018; Zhou et al., 2019). The data transfer is separated into a 1 Hz channel for all latitudes and a channel with higher instrument update rates for the area within  $\pm 65^{\circ}$  geocentric latitude. The 1 Hz channel is mainly used for housekeeping purposes and is not accessible for the CDSM team. The science phase of CSES is limited to  $\pm 65^{\circ}$  geocentric latitude because attitude control activities aboard the satellite over the polar regions cause too much magnetic interferences.

RC2\_3: On several places in the paper it is mentioned, that the phenomenon is still under investigation or similar. Are there some ideas already on the market (for example large local gradients or high frequent satellite signals)? What is already ruled out?

Authors' comments: Unfortunately nothing worth to publish.

RC2\_4: Page 5, line 106: These seven samples are averaged and serve as 1 Hz raw data of the CDSM instrument.: And how is the timestamp for this fraction calculated and set? Is it not a kind of challenge to align a patchy, spotlight-averaged value like this with other (presumably) continuously sampled and presumably averaged or filtered readings?

Changes proposed: P5 L105: ... those linked transients in the magnetic field strength read-out. Every second the mean value of these last seven samples is tagged with the time stamp of the fourth of last seven samples. The mean values These seven samples are averaged and serve as 1 Hz raw data of the CDSM instrument.

**RC2\_5: Line 164: I assume, all three vector magnetometers are already calibrated beforehand?**

Author's comments: The fluxgates have been calibrated beforehand.

**RC2\_6: Page 8, line 166, while the interference of FGM 1 is weak enough to be ignored: What is the threshold or criterion of being irrelevant?**

Changes proposed:

Fluxgates are inherently zero field detection devices where an artificial magnetic field is applied to cancel the environmental magnetic field in the sensor (Auster, 2008). For CSES this field can significantly influence the magnetic field measurement of the other sensors. The cross interferences were characterized with the sensors mounted on a dummy boom in a  $\mu$ -metal chamber (Zhou et al., 2018). The FGM 1 and FGM 2 sensors were located at the correct distances and orientation with respect to the CDSM position. The CDSM was replaced by a third fluxgate sensor for this test. Relevant for the CDSM is the feedback field generated by the FGM 2 sensor while the interference of FGM 1 is weak enough to be ignored. The influence of the FGM 2 feedback field  $F_{FG2}$   $F_{FG1}$  at the CDSM sensor position is

$$\begin{bmatrix} F_{FG2,x_{FG2}} \\ F_{FG2,y_{FG2}} \\ F_{FG2,z_{FG2}} \end{bmatrix} = I \begin{bmatrix} F_{x_{FG2}} \\ F_{y_{FG2}} \\ F_{z_{FG2}} \end{bmatrix} = 10^{-5} \begin{bmatrix} 5.34 & 1.97 & 0.67 \\ 1.33 & -7.82 & 0.00 \\ 0.00 & 1.90 & 2.76 \end{bmatrix} \begin{bmatrix} F_{x_{FG2}} \\ F_{y_{FG2}} \\ F_{z_{FG2}} \end{bmatrix}$$
$$\begin{bmatrix} F_{FG,x_{FG}} \\ F_{FG,x_{FG}} \\ F_{FG,x_{FG}} \\ F_{FG,x_{FG}} \\ F_{FG,x_{FG}} \end{bmatrix} = I \begin{bmatrix} F_{x_{FG}} \\ F_$$

where I is the matrix characteristic of the FGM 2 feedback field influence which depends on the Earth's magnetic field vector F. The FGM 2 sensor coordinates  $x_{FG2}$ ,  $y_{FG2}$ , and  $z_{FG2}$  fluxgate coordinates  $x_{FG7}$ ,  $y_{FG7}$ , and  $z_{FG2}$  fluxgate coordinates  $x_{FG7}$  is approx. the flight direction and  $z_{sat}$  points approx. to the center of Earth. The matrix characteristic of the FGM 1 feedback field influence is not available from ground tests. Nevertheless, with the fact of the same sensor design, orientation and given location, the influence of the FGM 1 feedback field  $F_{FG1}$  at the CDSM sensor position can be estimated as

$$\begin{bmatrix} F_{FG1, x_{FG2}} \\ F_{FG1, y_{FG2}} \\ F_{FG1, z_{FG2}} \end{bmatrix} \approx 0.11 \begin{bmatrix} F_{FG2, x_{FG2}} \\ F_{FG2, y_{FG2}} \\ F_{FG2, z_{FG2}} \end{bmatrix}$$

where  $F_{FG2}$  is the influence of the FGM 2 feedback field at the CDSM sensor position and  $x_{FG2}$ ,  $y_{FG2}$ , and  $z_{FG2}$  are the FG2 sensor coordinates. The CDSM scalar measurement is transformed into a vector as a function of F derived by FGM 2 and is corrected by  $F_{FG1}$  and  $F_{FG2}$ . In orbit the impact of the fluxgate sensors FGM 2 sensor is up to 4.4 nT 3.9 nT at the CDSM position and it depends on the magnetic field direction and strength. As an example the influence of the fluxgates FGM 2 during orbit segment 44270 is shown in Fig. 6. Changes proposed: Update of Fig. 6:

Figure 6: Example for the influence of the fluxgate feedback fields and the corresponding correction pattern.

**Changes proposed: Update of Fig. 8:**